# Assessing math anxiety in elementary schoolchildren through a Spanish version of the Scale for Early Mathematics Anxiety (SEMA)

**Noelia Sánchez-Pérez**[1]*, **Luis J. Fuentes**[2], **Carmen González-Salinas**[3]*

**1** Departamento de Psicología y Sociología, Facultad de Ciencias Sociales y Humanas, Universidad de Zaragoza, Teruel, Spain, **2** Departamento de Psicología Básica y Metodología, Facultad de Psicología, Universidad de Murcia, Murcia, Spain, **3** Departamento de Psicología Evolutiva y de la Educación, Facultad de Psicología, Universidad de Murcia, Murcia, Spain

* cgonzale@um.es (CGS); noeliasanchez@unizar.es (NSP)

**Data Availability Statement:** All relevant data are within the manuscript and its Supporting Information files.

## Abstract

Math anxiety (MA) affects students of all age groups. Because of its effects on children's academic development, the need to recognize its early manifestations has been highlighted. We designed a European-Spanish version of the Scale for Early Mathematics Anxiety (SEMA; Wu et al. (2012)), and assessed its psychometric properties in a sample of children aged 7 to 12 years. The participants (967 typically developing children) were elementary school students recruited from ten schools. Children reported their general and math anxiety levels in an individual session and performed nonverbal IQ and math abilities subtests in a group session. Teachers reported the final math grades. The psychometric indices obtained, and the resulting factor structure revealed that the European-Spanish version of the SEMA developed in this study is a reliable and valid measure to evaluate MA in children from 3rd to 6th grade. Moreover, we explored gender differences, that resulted in small effect sizes, which disappeared when controlling for trait anxiety. Differences across grades were found for both global MA and the numerical processing anxiety factor but not for the situational and performance anxiety factor. Finally, MA was negatively associated with students' math achievement, although the strength of the associations varied with the MA measure selected, the kind of math achievement analyzed, and the school stage considered. Our findings highlight the relevance of MA in elementary school and highlight the need for an early identification of students at risk of suffering MA to palliate the negative consequences of MA in children's cognitive and academic development.

## Introduction

Math anxiety (MA) is a worldwide problem affecting students of all age groups. Across Organization for Economic Co-operation and Development (OECD) countries, 59% of adolescents reported that they often worried about finding difficulties in mathematics classes; 33%

**Funding:** LFM: PSI2017-84556-P from the Spanish Ministry of Economy, Industry, and Competitiveness (FEDER funds) NSP: PID2019-107857GA-I00 from the Spanish Ministry of Science and Innovation. Website for both grants: https://www.ciencia.gob.es/portal/site/MICINN/ The funders had no role in study design, data collection and analysis, decision to publish, or preparation of the manuscript.

**Competing interests:** The authors have declared that no competing interests exist.

reported feeling very tense when facing mathematics homework; and 61% reported being concerned about getting poor grades in mathematics [1]. This situation appears to be even more severe in Spain, where 78% of students worry about poor grades in mathematics [2]. The level of anxiety experienced when faced with learning mathematics can affect students' academic trajectory, because individuals who suffer from higher MA also tend to exhibit poorer mathematics performance [1, 3–5], with negative consequences regarding future career choices [6].

Due to the relevance of MA and its effects on children's academic development, it has become apparent that there is a need to recognize its early manifestations in order to identify at-risk pupils and thus gain a full understanding of the MA phenomenon as well as its role in math achievement [7]. Aware of this need, we wanted to study MA at early educational levels, but the majority of the instruments available in the European-Spanish language were developed for high school and university populations. Given the dearth of reliable instruments to measure MA at the elementary school level in Spain, we set out to design a European-Spanish version of the Scale for Early Mathematics Anxiety (SEMA; [3]) and to test its psychometric properties in a sample of schoolchildren aged 7 to 12 years.

Math anxiety is defined as a "feeling of tension, apprehension or even dread, that interferes with the ordinary manipulation of numbers and the solving of mathematical problems" [8, p. 98]. Like any other phobia, MA affects individuals on three different levels: physiological reactions, cognitive effects, and avoidance behaviors [9]. Previous studies have found that schoolchildren [10] and adults [11] who scored high in MA showed greater physiological reactivity to mathematics, tended to experience more intrusive thoughts [12], and showed a behavioral disengagement bias specifically away from mathematical stimuli [13].

MA is not considered a separate diagnostic category in the main diagnostic systems for mental disorders–the DSM-V-R [14] and ICD-11 [15]–but it is included within general diagnostic labels, such as generalized anxiety disorder [16]. However, although MA and generalized anxiety were found to be positively correlated [3, 17, 18], the results were not consistent in early adolescence [19]. Furthermore, a meta-analysis with samples of children and young adults suggested that the two constructs are different [20].

Due to the overlap between general anxiety and MA, it would be expected that females would report higher rates of MA than males, given the extensive literature showing that females are significantly more likely than males to develop an anxiety disorder across the lifespan (see [21] for a review). Consistent with this expectation, Stoet et al.'s [22] study of high school students from 68 different nations participating in PISA, found that females reported higher levels of MA than males. Nonetheless, gender differences in elementary school are unclear: girls reported higher levels of MA than boys in some studies [23, 24], while no significant differences were found in others [25–27] (see [28] for more details). Regarding the study of MA across different age groups in elementary school, it has been suggested that mean levels of MA would decrease with advancing grades due to students' familiarization with the school tasks involved in math learning [29]. However, previous research yielded inconsistent results: some studies found a reduction in average MA scores across cohort years [30–32], yet others found no differences in MA across the school grades studied [3, 25, 33, 34]. Given the disparity in these results, further research is needed.

Regarding the academic consequences of MA, numerous studies have suggested that MA has a negative impact on math performance in all age groups: childhood [3, 5, 27, 31, 32], adolescence [4, 24, 35–37], and adulthood [38–41]. Nonetheless, not all individuals with high levels of MA have poor mathematics performance; multiple cognitive and contextual factors could affect the potential detrimental effects of MA on academic achievement (for a review, see [42]). Among these, the following factors have been identified: working memory [27, 43], metacognition [40], the type of problem-solving strategy used [31], parental involvement [44],

and parents' math anxiety [45]. Taken together, the above findings revealed that the MA-mathematics achievement link is quite complex, which underscores the need to better understand the nature of MA and the cognitive and contextual factors that may moderate or mediate the association with mathematics achievement.

In addressing the early manifestations of MA, a central issue concerns the measurement strategy selected. Since MA is an internal state of the individual, self-report has been the most commonly employed method of measurement. However, its use in children has been questioned (see [46]); due to their still immature metacognitive abilities, younger children would not be able to adequately report their feelings about MA. However, some studies have shown that children can understand the meaning of being nervous, anxious or tense about mathematics, and that they can reliably report their feelings [27, 33].

The key problem is, in fact, to generate a developmentally appropriate instrument that can allow children to reliably report on their inner states. Since MA was originally studied in adults, most authors put their efforts into adapting existing instruments to younger ages. As Ganley and McGraw [33] reviewed, these adaptations involved the following aspects: (1) The adaptation of the wording of the items as well as their content to the targeted grade level. As an example of this strategy, Suinn et al. [47] reviewed $4^{th}$ to $6^{th}$ grade mathematics teaching notebooks to generate items involving typical upper elementary school math calculations for the development of the MARS-E. Following the same strategy, some of the items in Wu et al.'s [3] SEMA referred to the second and third grade mathematics curricula used in the local geographical area. However, this grade specificity has been pointed out as a limitation because it does not to trace how MA changes with age [23]. (2) The use of a rating scale whose options have a clear meaning for the children. MA questionnaires typically use a Likert-type scale of 3 or 5 options, ranging from "not nervous at all" to "very very nervous". Especially for younger children, the inclusion of emoticons either replacing or illustrating each written option, has been deemed a successful strategy (e.g., [3, 25, 48]). (3) The adaptation of the length of questionnaires to avoid overloading children's cognitive abilities. Aware of this need, researchers have attempted to decrease the number of items while preserving the psychometric characteristics of the scales. A drawback of the resulting shortened questionnaires was that they might not capture the multifaceted nature of MA.

Indeed, it is recognized that MA is a multidimensional construct, and numerous studies have focused on identifying the different components of MA. In the adult population, the original MARS [49], and the subsequent abbreviated MARS-30 [50] have been the subject of multiple factorial studies, with different results found when different extraction methods, different samples, and different test samples have been used (see [51] for a review). Nonetheless, there is some consensus regarding the distinction of two subcomponents encompassed within an overall math anxiety: "mathematics test anxiety", which refers to anxiety associated with being tested in mathematics, or learning to take a mathematics tests, and "numerical anxiety", that refers to anxiety associated with manipulating numbers, basic arithmetic skills, and monetary decisions in everyday situations [51]. In children, there have been similar attempts to test the multidimensional nature of MA. Although factor solutions have varied depending on the underlying theoretical construct, the instrument, and the age period studied, the results have supported the view that multiple components can be identified early in life and that these components share a common core with those found in adolescents and adults [33]. More specifically, in early elementary school, Wu et al. [3] replicated the aforementioned structure of a global MA composed of two factors similar to the MARS and MARS-E [47, 49, 50], including a "numerical processing anxiety factor", that captures anxiety reactions related to doing mathematics-related work and problems, and the "situational and performance anxiety factor",

which refers to anxiety arising from social and examination situations requiring the use of mathematics.

The measurement of specific facets that are grouped together into higher-order factors is common practice in the area of personality. The appropriateness of using broad versus narrow measures in relation to criterion validity is still under debate (see [52] for a review). On the one hand, the broadband composite has higher reliability than the specific scales because the broadband factor contains more items than the specific ones and because the narrow scales are highly intercorrelated [53]. On the other hand, the meaning of the resulting composite score may be difficult to interpret and may lose predictive power because the association of certain subscales with the criterion of interest is diluted [54]. Aware of this issue, we aimed to test whether the factor structure of the SEMA found by Wu et al. [3] replicates in our sample. If it does, we should take into account the relative predictive power of the global MA compared to the numerical processing anxiety and the situational and performance anxiety factors separately.

In summary, the main objective of this study was to generate a European-Spanish adaptation of the Scale for Early Mathematics Anxiety (SEMA) and to assess its psychometric properties in a sample of typically developing children. Given that this instrument was designed for early elementary school grades, we also aimed to determine whether its application could be extended to subsequent grades, assessing children aged 7 to 12 years. If children with MA tend to react with apprehension or fear when faced with mathematical operations based on their previous experiences [46], they will negatively react even if the requested operations are apparently easy. Indeed, the items covering the numerical anxiety component of the MARS for adults involve simple arithmetic operations learned in elementary school, and the stimuli used in experiments with adults to induce anxiety reactions often involve undemanding tasks, such as the Numerical Stroop Task [55] or a two-digit addition verification task [56]. By adapting the SEMA to the European-Spanish language and childhood stage, we aimed to further analyze age and gender differences in children's MA, the relationship between MA and general anxiety, and the association of MA with mathematics performance.

## Materials and methods

### Participants

Participants were elementary school children recruited from ten schools collaborating in a broader project aimed at identifying individual and contextual factors associated with mathematics achievement in elementary school. The schools were located in rural or urban areas of the Region of Murcia (SE Spain). Initially, the sample was composed of 1126 children, but for validation purposes 159 participants (14.12%) were excluded because the child had a diagnosis of learning disability or clinical problems (n = 143), because the SEMA was not completed (n = 8), or because the family decided to drop out of the study (n = 6) or the school (n = 2). Consequently, a total of 967 children (483 boys, and 484 girls), aged 7 to 12 years (M = 9.36, SD = 1.26) participated. A questionnaire was administered to determine the sociodemographic characteristics of the families, but not all participants provided this information. The 71.45% of the sample reported their ethnic origin. Among these families, the majority were of European origin (95.2%), followed by Latin American (2%), African (.9%), Asian (.9%) and other (1%) origins; this distribution is representative of the ethnic variability in this geographical area. The 49.22% of the mothers reported their level of education; among them, 31.3% had an elementary school level of education, 20.4% had a high school level of education, and 48.3% had a university level of education. In the case of fathers, we received responses from 47.67% of the sample; among them, 38.2% completed elementary school, 23.2% completed high

school, and 38.6% completed university studies. The 72.29% of the families reported their family structure. In most of the families, children lived with both parents (83.7% of reported forms). Family income was reported by 63.29% of the sample, with 3.8% of the families reporting less than 750€ per month, 13% between 751 and 1200€ , 13.1% between 1201 and 1600€ , 16.2% between 1601 and 2000€ , 28.6% between 2001 and 3000€ , and 25.3% more than 3000€ . In 2018, the average monthly household income in the Region of Murcia was 1961.5€  [57].

## Instruments

**Math anxiety.**   Children completed a version of the SEMA [3] translated into Spanish and back-translated. They reported how anxious they felt if faced with a situation that required solving math questions (numerical processing anxiety factor, 10 items; e.g., "*Is this right*? *15– 7 = 8*") or when faced with social and exam situations involving mathematics (situational and performance anxiety factor, 10 items; e.g., "*You are about to take a math test*"). Following the authors' instructions, in an individual interview with a member of our staff, each written item was displayed separately while our staff member read it aloud. On another sheet, a Likert-type rating scale was displayed in words with 5 options, ranging from "not nervous at all" to "very very nervous"). Graded anxious faces illustrated each option to help children identify their anxiety levels (see S1 File,). The child was asked to point to the face that best represented how anxious s/he felt. The child was also invited to ask any questions related to the meaning of the items or the rating scale. The global MA and the subscale scores were calculated by summing the item scores, with higher scores indicating higher MA.

**Trait anxiety.**   Children reported their relatively stable characteristics (trait anxiety) using the Spanish version of the State-Trait Anxiety Inventory for Children (STAIC; [58]). For this study, we selected the 20 items referring to how they usually feel (trait anxiety). The estimated internal consistency in our sample was $\alpha = .80$.

**Math performance.**   To measure mathematics achievement, some researchers use teachers' reports, while others only administer standardized achievement tests. However, each measurement strategy has limitations; standardized tests may not cover the range of children's knowledge [59], while teachers' reports could be affected by subjective bias [60, 61]. This study overcame these limitations by including both measures: the ratings given by teachers–ranging from unsatisfactory (0) to outstanding (4)–and children's performance on standardized tests. More specifically, children completed the Calculation and Math fluency subtests of the Woodcock-Johnson III (WJ-III) Achievement battery ([51]; Spanish validation developed by Diamantopoulou et al. [62]). The Calculation subtest assesses the student's ability to perform simple mathematical calculations, including addition, subtraction, multiplication, and division, while the Math fluency subtest measures the ability to solve simple calculations quickly.

**Nonverbal IQ.**   Children responded to nonverbal IQ subtest of the Spanish version of the Kaufman Brief Intelligence Test (K-BIT) [63]. This measure assesses children's nonverbal reasoning and flexible problem-solving skills. The estimated internal consistency in our sample was $\alpha = .84$.

## Procedure

The study was approved by the Ethics Committee of the University of Murcia and it was conducted in accordance with the approved guidelines and the Declaration of Helsinki. In collaboration with the regional government, headteachers of the schools in the area of Murcia city and surroundings were informed about the project and invited to participate. Ten schools agreed to participate in the study. Letters were then sent to families describing the research project and included consent forms and questionnaires to collect sociodemographic

information. Once parents completed the forms, they returned them to the school. A member of the research team was available at the school to answer any questions or concerns raised by parents. All the measures were taken in the 2018–19 school year. From October to December 2018, children completed the SEMA, STAIC and another unpublished questionnaire related to MA in one-to-one sessions. In the second term (January-March 2019), children completed the nonverbal IQ (K-BIT) and math skills subtests in a group session (range = 6–24 children). Prior to each session, the child provided verbal consent. Finally, teachers provided the final mathematics grades obtained by children in June 2019.

## Results

### Normative data

First, the distribution of the item responses was analyzed to detect items that might not be sensitive to younger (3rd and 4th graders) or older students (5th and 6th graders). Stem-and-leaf plots revealed that there were 3 items with low variability (items 4, 6, and 12) for the younger and older students. In the case of the younger children, 80.89, 88.33, and 79.07% of the children replied that items 4, 6, and 12 respectively made them feel not nervous at all (option 1), whereas any of the other four alternative answers was only chosen by 19.11, 11.67, and 20.93% of the children respectively, with stem-and-leaf identifying those participants scoring higher than 1 as outliers. With respect to the older children, 91.28, 90.64, and 78.72% of them chose the option 1 for items 4, 6, and 12, respectively, whereas any of the other options for such items was chosen by 8.72, 9.36, and 21.28% of the children. Items 4 and 6 are part of the numerical processing anxiety scale but in contrast with the rest of the scale items, they do not involve mathematical calculations (see Table 1 for more details). The item 12 referred to doing homework; most of children in our sample reported not feeling nervous about performing math tasks at home. In addition, the 84.89 and 77.87% of just older children chose option 1 (not nervous at all) for items 2 and 13, respectively, whereas options higher than 1 were chosen by the 15.11 and 22.13% of older children for the same items, respectively. That is, solving one-digit additions (item 2) as well as counting one's savings (item 13) appear as cognitively undemanding tasks for older children. Although the shortening of the SEMA scales was not an aim of the present research, we excluded the aforementioned items from further analyses given that the identified floor effect indicated limited content validity. The removal of items with floor or ceiling effects has been proposed as a quality criterion for measurement properties of health status questionnaires [64].

Descriptive statistics, kurtosis and skewness coefficients for SEMA-related measures are shown in Table 1. Following SEMA's authors [3], global MA was calculated by adding the scores across all items. The mean total score for all the participants was 27.28 (possible range of 15–75), with a standard deviation of 7.81. ANOVA analyses showed that grade, $F(3,959) = 8.04$; $p < .001$; $\eta_p^2 = .025$, and gender, $F(1,959) = 7.04$; $p = .008$; $\eta_p^2 = .007$, were significant predictors, but the two factors did not interact, $F < 1$. These results indicated that grade and gender explained 2.5 and 0.7% of the variance in SEMA scores, respectively. Regarding the grade factor, post hoc comparisons with Bonferroni correction indicated that the third graders ($M = 29.15$, $SD = 8.54$) exhibited higher MA than the fifth ($M = 26.46$, $SD = 7.07$, $p = .001$) and sixth graders, ($M = 26.00$, $SD = 7.56$, $p < .001$). Regarding gender, the girls ($M = 27.92$, $SD = 8.08$) scored higher on MA than the boys ($M = 26.64$; $SD = 7.48$; *Cohen's d* = .16, indicating a small size effect).

Regarding the numerical processing anxiety scale, the results were similar to the MA global score, with significant effects for grade, $F(3,959) = 19.46$; $p < .001$; $\eta_p^2 = .057$, and gender, $F(1,959) = 7.15$; $p = .008$; $\eta_p^2 = .007$, but no interaction between the two factors, $F < 1$. The

**Table 1. Descriptive statistics for the SEMA items and variables.**

| | | | | | M (SD) | | | | | | |
|---|---|---|---|---|---|---|---|---|---|---|---|
| Item number | Original Item | Spanish Item | Kurtosis | Skew-ness | All sample | Boys (n = 483) | Girls (n = 484) | 3rd graders (n = 252) | 4th graders (n = 245) | 5th graders (n = 246) | 6th graders (n = 224) |
| 1 | George bought two pizzas that had six slices each. How many total slices did George have to share with his friends? | Jorge compró dos pizzas que tenían 6 porciones cada una. ¿Cuántas porciones tenía Jorge para compartir con sus amigos? | 1.70 | 1.23 | 1.78 (.86) | 1.76 (.85) | 1.79 (.87) | 1.99 (1.04) | 1.70 (.73) | 1.72 (.82) | 1.69 (.77) |
| 2* | Is this right? 9+7 = 18 | ¿Es esto correcto? 9 + 7 = 18 | 8.87 | 2.84 | .33 (.74) | .24 (.62) | .41 (.83) | .58 (1.01) | .31 (.66) | .22 (.57) | .17 (.51) |
| 3 | How much money does Annie have if she has two dimes and four pennies? | ¿Cuánto dinero tiene Ana si tiene dos monedas de diez céntimos y cuatro monedas de un céntimo? | 2.20 | 1.40 | 1.63 (.81) | 1.59 (.79) | 1.68 (.83) | 1.80 (.99) | 1.62 (.76) | 1.57 (.71) | 1.53 (.73) |
| 4* | How do you write the number *four hundred and eighty two*? | ¿Cómo escribirías el número cuatrocientos ochenta y dos? | 16.82 | 3.79 | .20(.59) | .21 (.57) | .20 (.62) | .37 (.83) | .21 (.53) | .11 (.39) | .12 (.47) |
| 5 | Draw an hour and minute hand on a clock so that it would read 3:15 PM | Dibuja un reloj con las manecillas de la hora y los minutos de modo que marque las 3:15 | 2.20 | 1.60 | 1.70 (1.00) | 1.78 (1.05) | 1.63 (.94) | 2.06 (1.23) | 1.79 (.97) | 1.53 (.79) | 1.39 (.79) |
| 6* | Draw a triangle and a square on the board | Dibuja un cuadrado y un triángulo en la pizarra. | 16.97 | 3.89 | .15 (.51) | .19 (.58) | .11 (.42) | .17 (.60) | .18 (.52) | .15 (.45) | .11 (.43) |
| 7 | Count aloud by 5 s from 10 to 55 | Cuenta en voz alta de 5 en 5 desde 10 hasta 55. | 4.43 | 1.92 | 1.54 (.81) | 1.48 (.77) | 1.59 (.85) | 1.56 (.86) | 1.65 (.86) | 1.54 (.74) | 1.39 (.75) |
| 8 | What time will it be in 20 min? | ¿Qué hora será dentro de 20 minutos? | 2.32 | 1.56 | 1.72 (.96) | 1.61 (.89) | 1.83 (1.01) | 2.07 (1.12) | 1.75 (.95) | 1.57 (.85) | 1.46 (.75) |
| 9 | Is this right?15−7 = 8? | ¿Es esto correcto? 15–7 = 8. | 4.37 | 1.90 | 1.44 (.70) | 1.34 (.60) | 1.53 (.78) | 1.63 (.89) | 1.40 (.64) | 1.35 (.57) | 1.35 (.61) |
| 10 | Daisy has more money than Ernie. Ernie has more money than Francesca. Who has more money–Daisy or Francesca? | Marta tiene más dinero que Quique. Quique tiene más dinero que Diana. ¿Quién tiene más dinero Marta o Diana? | .77 | 1.06 | 1.95 (.99) | 1.89 (.97) | 2.00 (1.01) | 1.98 (1.04) | 1.93 (1.02) | 1.89 (.96) | 1.99 (.95) |
| 11 | You are in math class and your teacher is about to teach something new | Estás en clase de mates y tu profesor va a explicar algo nuevo. | 3.27 | 1.70 | 1.63 (.86) | 1.63 (.87) | 1.63 (.86) | 1.75 (1.01) | 1.63 (.86) | 1.57 (75) | 1.56 (.80) |
| 12* | You have to sit down to start your math homework | Tienes que sentarte para comenzar tus deberes de mates. | 11.41 | 3.11 | .29 (.68) | .33 (.73) | .25 (.64) | .35 (.79) | .27 (.67) | .28 (.64) | .27 (.62) |
| 13* | You are adding up all the money in your piggy bank | Estás calculando el dinero que tienes en tu hucha. | 6.11 | 2.31 | .43 (.78) | .43 (.80) | .42 (.76) | .62 (.94) | .44 (.71) | .30 (.62) | .33 (.78) |
| 14 | Someone asked you to cut up an apple pie into four equal parts | Alguien te pide que cortes la tarta de manzana en cuatro partes iguales. | 2.00 | 1.42 | 1.75 (.93) | 1.83 (.95) | 1.67 (.91) | 1.90 (1.09) | 1.80 (.85) | 1.63 (.84) | 1.65 (.90) |
| 15 | You are about to take a math test | Estás a punto de hacer un examen de matemáticas. | -.71 | .41 | 2.78 (1.22) | 2.62 (1.19) | 2.94 (1.23) | 2.73 (1.39) | 2.79 (1.22) | 2.74 (1.12) | 2.87 (1.11) |

(*Continued*)

**Table 1.** (Continued)

| Item number | Original Item | Spanish Item | Kurtosis | Skew-ness | M (SD) | | | | | | |
|---|---|---|---|---|---|---|---|---|---|---|---|
| | | | | | All sample | Boys (n = 483) | Girls (n = 484) | 3rd graders (n = 252) | 4th graders (n = 245) | 5th graders (n = 246) | 6th graders (n = 224) |
| 16 | You are in math class and you do not understand something. You ask your teacher to help you | Estás en clase de mates y no comprendes algo. Preguntas al profesor. | 1.43 | 1.29 | 1.84 (.97) | 1.82 (.99) | 1.86 (.95) | 1.74 (1.01) | 1.84 (.97) | 1.87 (.95) | 1.93 (.95) |
| 17 | Your teacher gives you a bunch of addition problems to work on | Tu profesor te da un montón de problemas de sumas para que las hagas. | .61 | 1.19 | 1.96 (1.16) | 1.95 (1.19) | 1.98 (1.14) | 2.18 (1.28) | 1.96 (1.16) | 1.92 (1.09) | 1.79 (1.05) |
| 18 | Your teacher gives you a bunch of subtraction problems to work on | Tu profesor te da un montón problemas de restas para que las hagas | .95 | 1.26 | 1.88 (1.08) | 1.81 (1.06) | 1.96 (1.10) | 2.22 (1.27) | 1.87 (1.06) | 1.77 (.99) | 1.64 (.85) |
| 19 | You are in class doing a math problem on the board | Estás en clase resolviendo un problema de mates en la pizarra. | .24 | .95 | 2.25 (1.14) | 2.13 (1.12) | 2.36 (1.16) | 2.13 (1.22) | 2.20 (1.09) | 2.33 (1.08) | 2.35 (1.16) |
| 20 | You are listening as your teacher explains to you how to do a math problem | Escuchas a tu profesor explicándote cómo hacer un problema de mates. | 5.06 | 2.08 | 1.43 (.74) | 1.40 (.67) | 1.46 (.80) | 1.41 (.79) | 1.42 (.72) | 1.46 (.69) | 1.43 (.76) |
| | Numerical Processing Anxiety factor | | 1.98 | 1.27 | 11.76 (3.68) | 11.46 (3.40) | 12.05 (3.92) | 13.09 (4.27) | 11.85 (3.42) | 11.18 (3.10) | 10.79 (3.40) |
| | Situational and Performance Anxiety factor | | .73 | .90 | 15.52 (4.99) | 15.18 (4.89) | 15.86 (5.06) | 16.06 (5.42) | 15.49 (4.85) | 15.28 (4.70) | 15.21 (4.91) |
| | Global MA | | 1.17 | .99 | 27.28 (7.81) | 26.64 (7.48) | 27.92 (8.08) | 29.15 (8.54) | 27.34 (7.62) | 26.46 (7.07) | 26.00 (7.56) |

Original items were obtained from Wu et al., (2012): items 1 to 10 constitute the Numerical Processing Anxiety factor; items 11 to 20 belong to the Situational and Performance Anxiety factor. Items with the asterisk (2,4,6, 12, and 13) were removed from further analyses.

results indicated that grade and gender explained 5.7 and 0.7 of the numerical processing anxiety factor, respectively. For the grade factor, the analysis indicated that the third graders ($M$ = 13.09, $SD$ = 4.27) exhibited higher MA than the fourth ($M$ = 11.85, $SD$ = 3.42, $p$ = .001), fifth ($M$ = 11.18, $SD$ = 3.10, $p < .001$), and sixth graders ($M$ = 10.79, $SD$ = 3.40, $p < .001$). Moreover, the fourth graders also reported higher MA than the sixth graders ($p$ = .008). Regarding gender, the girls ($M$ = 12.05, $SD$ = 3.92) scored higher in MA than the boys ($M$ = 11.46, $SD$ = 3.40; with $Cohen's\ d$ = .16, meaning a small size effect.

Regarding the situational and performance anxiety factor, the ANOVA revealed that there was neither a significant main effect of grade, $F(3,959)$ = 1.52; $p$ = .207, $\eta_p^2$ = .005 nor a grade by gender interaction, $F < 1$. However, the main effect of gender was significant, $F(1,959)$ = 4.79; $p$ = .029; $\eta_p^2$ = .005. The results revealed that gender explained 0.5% of the subscale scores, with the girls ($M$ = 15.86, $SD$ = 5.06) scoring higher in MA than the boys ($M$ = 15.18, $SD$ = 4.89; Cohen's $d$ = .14, indicating also a small size effect). Finally, following the suggestion of Ganley and McGraw [33], trait anxiety was introduced as a covariate. ANCOVAs indicated that there were significant main effects of grade for global MA, $F(3,957)$ = 9.64; $p < .001$; $\eta_p^2$ = .029, and for the numerical processing anxiety factor, $F(3,957)$ = 22.44; $p < .001$; $\eta_p^2$ = .066, but not for the situational and performance anxiety factor, $F(3,957)$ = 1.54; $p$ = .175; $\eta_p^2$ = .002. In the case of gender, there were no significant differences in global MA, $F(3,957)$ = 3.40; $p$ = .066; $\eta_p^2$ = .004, either for the numerical processing anxiety factor, $F(3,957)$ = 3.83; $p$ = .051;

$\eta_p^2 = .004$, or for the situational and performance anxiety factor, $F(3,957) = 1.85$; $p = .175$; $\eta_p^2 = .002$. The grade by gender interaction was not significant for the global MA, numerical processing anxiety factor, or situational and performance anxiety factor (all $Fs < 1$). To summarize, grade effects were found in relation to math anxiety (global MA and the numerical processing anxiety factor), even when trait anxiety was introduced as a covariate. On the other hand, gender effects were found to be significant in predicting math anxiety (global MA, numerical processing anxiety factor and situational and performance anxiety factor), but these effects were small and disappeared when trait anxiety was considered in the analyses. Interestingly, the trait anxiety variable in our sample exhibited gender effects, $t(964) = 2.09$, $p = .037$, with the girls ($M = 33.67$, $SD = 6.45$) reporting feeling more anxious than the boys ($M = 32.83$, $SD = 6.08$; *Cohen's d* = .13, meaning a small size effect).

## Reliability

Internal consistency as measured by Cronbach's α, was .83 for the global MA, with item-total correlations ranging from .33 to .55 (average of .45). The Spearman–Brown coefficient was computed to measure the split-half reliability, with a result of .79. In the case of the numerical processing anxiety and situational and performance anxiety scales, internal consistency also reached an adequate level (α = .70, and .76), with item-total correlations ranging from .34 to .47, and from .33 to .54, respectively. These results were comparable to the reliability indices reported in the original study [3], in which the authors found an internal consistency of .87, .80, and .77 for global MA, numerical processing anxiety, and situational and performance anxiety scales, respectively (see [3] for details). Note that the SEMA-Spanish version is composed of fewer items than the original version. Moreover, the internal consistency of the global MA, numerical processing anxiety, and situational and performance anxiety scales was also calculated for the younger (α = .82, .68, and .74), and older students (α = .84, .69, and .77). Item-total correlations ranged from .28 to .55, .30 to .46, and .30 to .54 for the younger students, and from .35 to .60, .36 to .48, and .33 to .57 for the older students, for the global MA, numerical processing anxiety, and situational and performance anxiety scales, respectively.

## Structural validity

To test the two-factor solution proposed for the SEMA questionnaire [3], confirmatory factor analysis (CFA) was run with the RStudio program [65]. The estimator method unweighted least squares (ULS) with a polychoric correlation matrix was used, as it provides accurate results for ordinal variables [66].

The two-factor model yielded good fit indices: CFI was .97; TLI was .96; RMSEA and SRMR were .06 and .06, respectively. Also, the numerical processing anxiety scale and the situational and performance anxiety scale had standardized factor loadings from .47 to .63 and from .39 to .65, respectively, and all items loaded significantly, $p < .001$.

Moreover, four additional CFAs were separately computed; two split the sample by gender (one CFA for girls, another for boys) and another two split the sample by school stage, including younger (3rd and 4th graders), and older students (5th and 6th graders). The goodness-of-fit indices were adequate for all the CFA models (see Table 2), providing support for the two-factor solution of SEMA.

Since the two factors were highly correlated ($r = .84$), a unifactorial model for the SEMA questionnaire was subsequently tested. The analysis yielded adequate fit indices: CFI was .96; TLI was .96; RMSEA and SRMR were .06 and .06, respectively, and all items loaded significantly, $p < .001$. Fit indices for the one-factor model matched those of the two-factor model

**Table 2. Goodness of fit indices for the one- and two-factor models of SEMA split by gender (girls and boys) and age, with younger (3rd and 4th graders), and older students (5th and 6th graders).**

| | One-factor model | | | | Two-factor model | | | |
|---|---|---|---|---|---|---|---|---|
| Sample | CFI | TLI | RMSEA | SRMR | CFI | TLI | RMSEA | SRMR |
| Girls | .96 | .95 | .06 | .06 | .97 | .96 | .05 | .06 |
| Boys | .98 | .98 | .04 | .05 | .98 | .98 | .03 | .05 |
| Younger students | .97 | .96 | .05 | .05 | .97 | .96 | .05 | .05 |
| Older students | .98 | .97 | .03 | .05 | .99 | .99 | .02 | .05 |

except for the CFI index. Moreover, the one-factor model was also tested for both the younger and older students as well as for the girls and the boys (see Table 2).

## Math anxiety and its relationship to trait anxiety

We analyzed whether MA was related to trait anxiety measured by the STAIC [58]. Zero-order correlations revealed that trait anxiety was significantly correlated with global MA, $r = .50$, $p < .001$, the numerical processing anxiety factor, $r = .42$, $p < .001$, and the situational and performance anxiety factor, $r = .47$, $p < .001$.

## Math anxiety and its relationship to math grades

We examined whether MA predicted math performance as measured by math grades, while taking into account potential control variables.

The first step was to analyze whether math grades were related to our potential control variables (gender, grade, nonverbal IQ, and trait anxiety). We found a significant correlation with nonverbal IQ, $r = .33$, $p < .001$, and trait anxiety, $r = -.17$, $p < .001$. Moreover, the ANOVA analysis yielded statistically significant results, with children enrolled at 3rd grade obtaining higher scores ($M = 7.87$, $SD = 1.53$) than 4th graders ($M = 7.43$, $SD = 1.68$), $F(3,956) = 3.48$, $p = .019$. Consequently, these variables were considered in subsequent analyses.

The second step was to assess the correlation between MA and math grades. The results showed that global MA, the numerical processing anxiety factor, and the situational and performance anxiety factor were significantly and negatively related to math grades, $r = -.26$, $p < .001$, $r = -.22$, $p < .001$, and $r = -.24$, $p < .001$, respectively.

Finally, we computed two stepwise regressions to predict math grades. Trait anxiety, grade, and nonverbal IQ were introduced in the first step, and global MA (first model) and numerical processing anxiety and situational and performance anxiety factors (second model) in the second step. In the first step of the MA global regression, $F(3,925) = 54.61$, $p < .001$, $R^2_{adj} = .148$, the results showed that trait anxiety, $\beta^\wedge = -.15$, $p < .001$, grade, $\beta^\wedge = -.15$, $p < .001$, and nonverbal IQ, $\beta^\wedge = .36$, $p < .001$, were significant predictors of children's math grades. The second step, $F(4,924) = 53.06$, $p < .001$, $R^2_{adj} = .183$, indicated that global MA was a significant predictor of children's math grades, $\beta^\wedge = -.23$, $p < .001$, as well as grade, $\beta^\wedge = -.18$, $p < .001$, and nonverbal IQ, $\beta^\wedge = .34$, $p < .001$, but not trait anxiety, $\beta^\wedge = -.04$, $p = .266$. In the second regression, numerical processing anxiety and situational and performance anxiety factors were introduced as independent variables to analyze the contribution of each factor to math grades. The first step regression, $F(3,925) = 54.61$, $p < .001$, $R^2_{adj} = .148$, revealed that there were a significant contribution of trait anxiety, $\beta^\wedge = -.15$, $p < .001$, grade, $\beta^\wedge = -.15$, $p < .001$, and nonverbal IQ, $\beta^\wedge = .36$, $p < .001$; whereas the step two results, $F(5,923) = 42.45$, $p < .001$, $R^2_{adj} = .183$, showed that the numerical processing, $\beta^\wedge = -.12$, $p = .002$, and the situational and performance anxiety factors, $\beta^\wedge = -.13$, $p = .001$, were significant predictors of children's math grades, as well as

**Table 3. Zero-order correlations between WJ-III scale scores and potential control variables (gender, grade, nonverbal IQ, and trait anxiety).**

| WJ-III scales | Potential control variables | | | |
| --- | --- | --- | --- | --- |
| | Gender | Grade | Nonverbal IQ | Trait anxiety |
| Calculation | .05 | .44*** | .73*** | -.08** |
| Math fluency | .21*** | .51*** | .41*** | -.13*** |

Gender was coded as 0 for girls and 1 for boys

***$p < .001$

** $p < .01$.

grade, $\hat{\beta} = -.18$, $p < .001$, and nonverbal IQ, $\hat{\beta} = .34$, $p < .001$, but not trait anxiety, $\hat{\beta} = -.04$, $p = .266$.

## Math anxiety and its relationship to standardized math achievement test scores

Following the steps previously described, we examined whether MA predicted math performance, measured by the Calculation and Math fluency subtests from the Spanish version [62] of the WJ-III battery [67].

The first step was to analyze whether standardized math test scores were associated with potential control variables (gender, grade, nonverbal IQ, and trait anxiety). The results are presented in Table 3.

The next step was to study the correlation between MA and math test scores. The results showed that global MA, the numerical processing anxiety factor, and the situational and performance anxiety factor were significantly and negatively related to all the math test scores (see Table 4).

The last step was to compute four hierarchical regressions to predict math test scores (Calculation, and Math fluency): two considering global MA as an independent variable and two other regressions with numerical processing anxiety and situational and performance anxiety factors as independent variables. In the first set of regressions, the global MA was a significant predictor of all the measures of math performance, even taking into account the control variables (see Table 5). In the second set of regressions, the results revealed that the numerical processing anxiety factor was a significant predictor of Math fluency after including the control variables, but it did not predict calculation skills (see Table 6). The situational and performance anxiety factor did not yield a significant contribution to math abilities.

To determine whether the weaker relationship between the situational and performance anxiety factors and the standard tests that were observed in the regression models compared with the correlation analyses, were caused by either the correlation between the two factors or the inclusion of the covariates in the model, we performed an additional regression analysis. As shown in Table 7, the numerical processing anxiety factor did yield a significant contribution to Calculation and Math fluency abilities, but the situational and performance anxiety factor did not.

**Table 4. Zero-order correlation between math anxiety and WJ-III scale scores.**

| SEMA scales | WJ-III scales | |
| --- | --- | --- |
| | Calculation | Math fluency |
| Global MA | -.21*** | -.29*** |
| Numerical processing anxiety factor | -.23*** | -.33*** |
| Situational and performance anxiety factor | -.15** | -.21*** |

**Table 5. Hierarchical regressions predicting WJ-III scale scores with global MA as an independent variable.**

| | | | | | IV: global MA | | | |
| --- | --- | --- | --- | --- | --- | --- | --- | --- |
| | | | | | Model 1 | | Model 2 | |
| DV: Calculation scale | F | ΔF | $R^2_{adj}$ | $\Delta R^2_{adj}$ | Standardized β | p | Standardized β | p |
| Model | 443.93*** | -104.51*** | .58 | .004 | | | | |
| Grade | | | | | .26 | < .001 | .25 | < .001 |
| Nonverbal IQ | | | | | .66 | < .001 | .65 | < .001 |
| Trait anxiety | | | | | -.04 | .063 | .02 | .936 |
| Global MA | | | | | | | -.13 | < .001 |
| DV: Fluency scale | F | ΔF | $R^2_{adj}$ | $\Delta R^2_{adj}$ | Standardized β | p | Standardized β | p |
| Model | 131.47*** | -18.01*** | .37 | .017 | | | | |
| Gender | | | | | .21 | < .001 | .21 | < .001 |
| Grade | | | | | .42 | < .001 | .41 | < .001 |
| Nonverbal IQ | | | | | .24 | < .001 | .22 | < .001 |
| Trait anxiety | | | | | -.09 | < .001 | -.02 | .556 |
| Global MA | | | | | | | -.16 | < .001 |

***p <. 001.

## Math anxiety and its relevance for math performance by school stage

In the last step, we aimed to study whether there was an interaction between MA and school stage in predicting math performance. With that aim, we split the sample into younger (3rd and 4th graders) versus older children (5th and 6th graders), and ran the previous hierarchical regressions with the same covariates to predict math performance (math grades, and Calculation and Math fluency subtests), and introduced the interaction term between school stage and MA (as a centered variable). The analyses involving global MA yielded a significant

**Table 6. Hierarchical regressions predicting WJ-III scale scores with numerical processing anxiety and situational and performance anxiety factors as independent variables.**

| | | | | | IV: Numerical processing anxiety and situational and performance anxiety factors | | | |
| --- | --- | --- | --- | --- | --- | --- | --- | --- |
| | | | | | Model 1 | | Model 2 | |
| DV: Calculation scale | F | ΔF | $R^2_{adj}$ | $\Delta R^2_{adj}$ | Standardized β | p | Standardized β | p |
| Model | 443.93*** | -172.68*** | .59 | .004 | | | | |
| Grade | | | | | .26 | < .001 | .25 | < .001 |
| Nonverbal IQ | | | | | .66 | < .001 | .65 | < .001 |
| Trait anxiety | | | | | -.04 | .063 | -.00 | .936 |
| Numerical processing anxiety factor | | | | | | | -.05 | .127 |
| Situational and performance anxiety factor | | | | | | | -.05 | .077 |
| DV: Fluency scale | F | ΔF | $R^2_{adj}$ | $\Delta R^2_{adj}$ | Standardized β | p | Standardized β | p |
| Model | 131.47*** | -36.58*** | .38 | .018 | | | | |
| Gender | | | | | .21 | < .001 | .21 | < .001 |
| Grade | | | | | .42 | < .001 | .40 | < .001 |
| Nonverbal IQ | | | | | .23 | < .001 | .22 | < .001 |
| Trait anxiety | | | | | -.09 | < .001 | -.02 | .583 |
| Numerical processing anxiety factor | | | | | | | -.13 | < .001 |
| Situational and performance anxiety factor | | | | | | | -.05 | .119 |

***p < .001.

**Table 7. Regressions analyses predicting WJ-III scale scores with numerical processing anxiety and situational and performance anxiety factors as independent variables.**

| DV: Calculation scale | F | $R^2_{adj}$ | Standardized β | p |
|---|---|---|---|---|
| Model | 26.41*** | .05 | | |
| Numerical processing anxiety factor | | | -.22 | < .001 |
| Situational and performance anxiety factor | | | -.02 | .706 |
| DV: Fluency scale | F | $R^2_{adj}$ | Standardized β | p |
| Model | 54.29*** | .11 | | |
| Numerical processing anxiety factor | | | -.32 | < .001 |
| Situational and performance anxiety factor | | | -.01 | .786 |

interaction effect in predicting math grades ($F(5,923)$ = 41.04, $p < .001$, $R^2_{adj}$ = .177; $\beta^\wedge$ = .18, $p$ = .048) but not Calculation ($F(5,923)$ = 273.92, $p < .001$, $R^2_{adj}$ = .595; $\beta^\wedge$ = -.021, $p$ = .747) or Math fluency ($F(6,896)$ = 82.513, $p < .001$, $R^2_{adj}$ = .352; $\beta^\wedge$ = .003, $p$ = .970) subtest scores. As shown in Fig 1, higher MA levels were associated with lower math grades for both younger and older children; however, the effects of MA on children's math grades tended to be stronger for the younger children than for the older children. In the case of the SEMA subscales, none of the interaction terms between numerical processing anxiety and school stage was significant in predicting math grades ($F(5,923)$ = 37.68, $p < .001$, $R^2_{adj}$ = .165; $\beta^\wedge$ = ..05, $p$ = .213), Calculation ($F(5,923)$ = 272.519, $p < .001$, $R^2_{adj}$ = .594; $\beta^\wedge$ = -.007, $p$ = .920), or Math fluency ($F(6,896)$ = 83.51, $p < .001$, $R^2_{adj}$ = .354; $\beta^\wedge$ = -.051, $p$ = .540) subtest scores. However, a significant interaction was found between situational and performance anxiety and school stage in predicting math grades ($F(5,923)$ = 39.44, $p < .001$, $R^2_{adj}$ = .172; $\beta^\wedge$ = .20, $p$ = .032) but not for Calculation ($F(5,923)$ = 272.39, $p < .001$, $R^2_{adj}$ = .594; $\beta^\wedge$ = -.029, $p$ = .657) or Math fluency ($F(6,896)$ = 79.24, $p < .001$, $R^2_{adj}$ = .342; $\beta^\wedge$ = .033, $p$ = .697) subtest scores, which was similar to the global MA results. Similar to the significant interaction between global MA and school stage in predicting math grades, higher scores on the situational and performance anxiety factor predicted lower math grades for younger and older children, but this negative association tended to be stronger for the younger than for the older children (see Fig 2).

## Discussion

In this study, a European-Spanish version of the SEMA [3] was designed, and its psychometric characteristics were analyzed in a representative population-based sample of children aged 7 to 12 years, including a wide range of socioeconomic backgrounds and a balanced gender distribution. Since this instrument was designed for the early grades of elementary school, we also examined whether its application could be extended to subsequent grades. Adapting this instrument to the European-Spanish language and childhood stage, we explored the structure of MA in this age period and analyzed age and gender differences in children's self-reported MA, the relationship between MA and general anxiety, and the association of MA with mathematics achievement.

### Factorial structure and reliability of the Spanish SEMA

Our results revealed that the Spanish version of the SEMA questionnaire is a reliable and construct-valid instrument for measuring MA at school age. Internal consistency indices proved to be adequate for the total sample, as well as for younger and older children. Regarding the structure of MA, we replicated the original two-factor structure (numerical processing anxiety and situational and performance anxiety factors) proposed by Wu and colleagues [3]. As

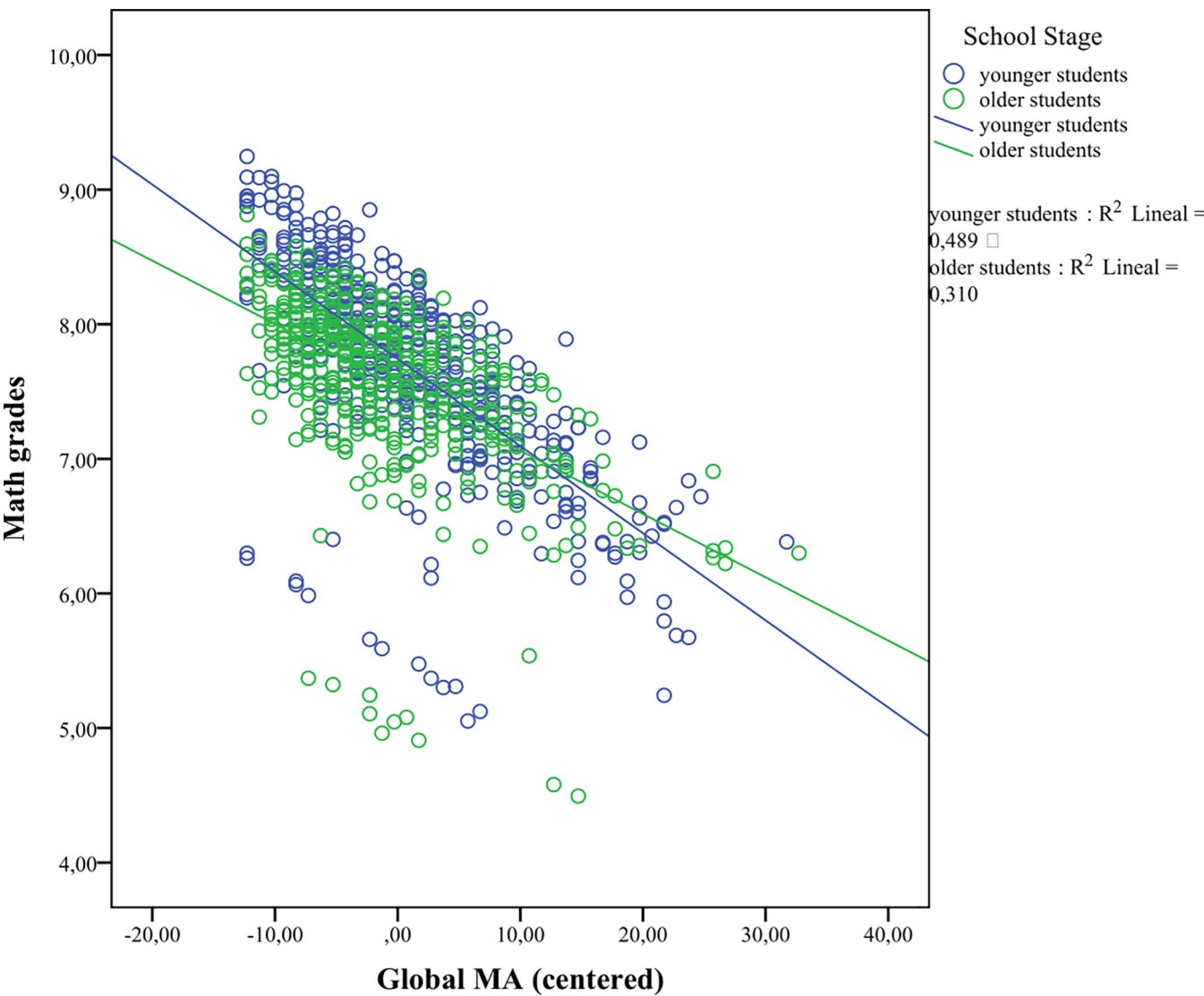

**Fig 1. Interaction term between school stage and MA (as a centered variable) in predicting math grades.**

mentioned above, the SEMA is a standardized measure originally designed to examine MA in children aged 7 to 9 years. Interestingly, in our sample, this factorial structure seemed appropriate for both younger and older children. Moreover, the two-factor structure was replicated for both boys and girls. In line with previous studies that also used questionnaires derived from the MARS [47, 50, 68, 69], a component of negative feelings when handling numbers (e.g., performing an addition) is distinguished from the situations in which math operations are required (e.g., classwork, exams, etc.). Nonetheless, the high correlation between the numerical processing anxiety and situational and performance anxiety factors found in our sample, and the good fit indices reached for the one-factor model also lend support to a broad global MA measure constituted by the aforementioned scales. Similarly, Baloğlu and Balgalmis [70], and Roick et al. [71] found strong correlations between the extracted factors from their respective adaptations of the MARS-E, a fact that has led to the interpretation that both the MARS and other questionnaires derived from it imply a single dimension of MA, i.e., the

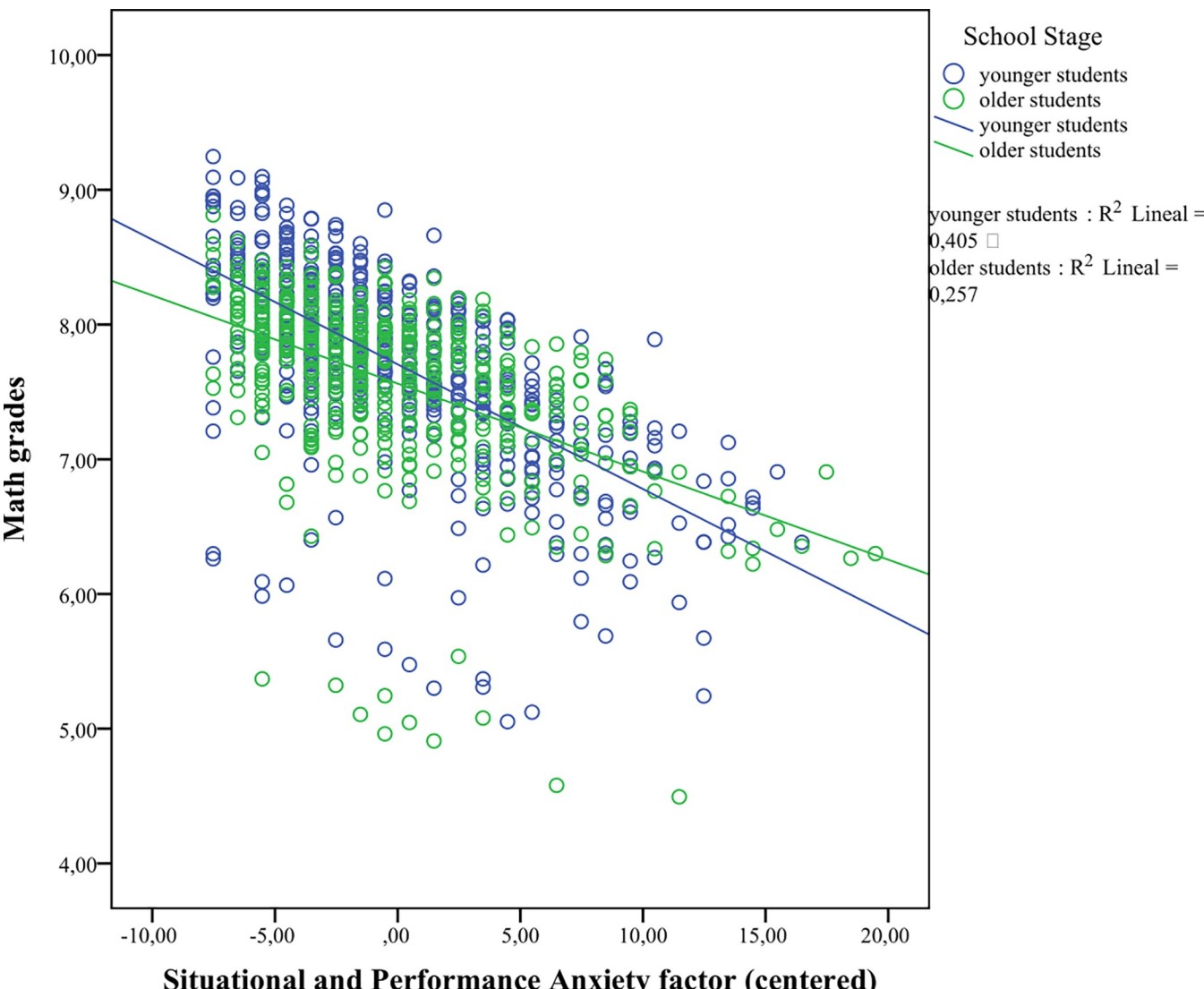

**Fig 2. Significant interaction between school stage and situational and performance anxiety factor (as a centered variable) in predicting math grades.**

affective component [72]. From a different approach, Henschel and Roick [72] included a cognitive component characterized by conscious worry or concern. In addition, another promising effort has been made by Kohn et al. (2013), with the design of an instrument aimed at identifying the four components of anxiety, i.e., affective, cognitive, behavioral, and physiological, that can be elicited in different mathematics-related situations.

## Age and gender differences

We found significant age differences for global MA and the numerical processing anxiety factor between younger children and their older counterparts. Although children might react less negatively to math operations as a consequence of familiarity with school tasks throughout the elementary school years [29], it is likely that these results are due to an effect of the different levels of difficulty of the items comprising the numerical processing anxiety factor for younger versus older children. In fact, Wu et al. [3] based the content of the items on the second- and

third-grade mathematics curricula obtained from the local geographical area, which could have produced a floor effect on the levels of anxiety experienced by the older children. In contrast, the situational and performance anxiety factor did not show differences by age, i.e., the levels of anxiety produced by situations such as attending to teacher explanations or taking a math test remained stable across ages, and this result replicates previous findings [3, 25, 33, 34].

Regarding gender differences, the girls scored slightly higher than the boys on the global MA as well as on the numerical processing anxiety and situational and performance anxiety factors, in line with previous studies [23, 24]. However, when the general levels of anxiety were controlled for, the differences found for global MA, the numerical processing, and situational and performance anxiety factors disappeared. Similarly, Ganley and McGraw [33] found gender differences in the levels of MA reported by the children through the MASY-C questionnaire, which disappeared when general anxiety was used as a covariate. Szczygiel's [73] mediational study in a sample of early elementary school children may help to explain our results, as she found that gender determined levels of general anxiety, which in turn strongly predicted MA. In conclusion, our findings support the notion that gender differences in MA in the elementary school years are small and could be explained by the fact that girls tend to experience higher levels of general anxiety.

**Math anxiety and its relationship to general anxiety.** Regarding the relationship between MA and general anxiety, our findings suggest that the children who scored high in MA also tended to report higher general anxiety, which is consistent with previous research [3, 17, 18]. Following Malanchini et al. [57], the overlap between the two types of anxiety is consistent with the research suggesting partial overlap in the cognitive [74] and brain [55] processes that are involved in both general anxiety and MA. On the other hand, although both types of anxiety are related, their level of association leads us to conclude that they are different constructs, which is consistent with the interpretation of a previous meta-analysis [20]. Moreover, the fact that trait anxiety ceased to significantly contribute when MA was introduced into the regression models for predicting performance, lends further support to the differentiation between these two constructs. In the same vein, genetics studies have suggested that although some genetic and environmental factors contribute to various measures of anxiety (including general anxiety and MA), a substantial part of the nonshared genetic and environmental influences are specific to each anxiety construct [75].

**Math anxiety and math performance.** The contribution of MA to math performance was analyzed taking into account teachers' reports and standardized achievement tests, and different results were found depending on the MA measure selected (broad versus narrow) and the kind of math performance considered. Global MA appears to be a robust measure that proved its contribution to all math performance measures, with higher levels of tension or apprehension in coping with math learning associated with poorer math performance as measured via teachers' reports or standard achievement tests. This finding is in consonance with the previous literature reviewed [3, 5, 27, 31, 32] and gives support to the SEMA as a valid instrument to identify MA in elementary school students.

The subscales also appeared as significant predictors of math achievement, but showed a weaker contribution that varied as a function of the specific math achievement index analyzed. On the one hand, the situational and performance anxiety factor only predicted children's grades given by teachers. Teachers' assessments are usually obtained in the context of the classroom, and high levels of anxiety experienced by children in daily situations (such as performing class activities, asking the teacher for help, solving a problem on the board, or taking an exam) could have produced a detrimental effect on their performance. Additionally, given that individuals with greater MA also exhibit behavioral disengagement from mathematical stimuli

[13], teachers might have taken that behavior into account, resulting in a lower grade in mathematics.

On the other hand, the numerical processing anxiety factor negatively predicted math grades as well as children's ability to solve simple calculations quickly, as measured through standardized tests, in line with previous studies [3]. In our sample, the children who reported feeling more anxious regarding handling math operations tended to perform math class assignments and exams more poorly, and respond slower and less accurately when performing simple arithmetic calculations. Previous research pointed to the presence of high physiological reactivity to mathematics and intrusive thoughts as pathways that connect MA and math performance [10–12]. Interestingly, the numerical processing anxiety factor did not predict Calculation scores whereas other factors, especially non-verbal intelligence, showed a more relevant role in explaining children's differences in arithmetic abilities. In our sample, children's negative feelings to mathematics seem to interfere with speed rather than accuracy when arithmetic operations are performed.

However, it is necessary to be cautious with respect to the above interpretation. Given that the design of this study was correlational, we could alternatively interpret MA as the resulting emotion that low math performers feel when facing math operations, as found by Ma and Xu [76] in a sample of middle and high school students. However, a reciprocal model of MA and math performance has been suggested as the most plausible model [77].

In conclusion, whereas global MA represents a robust measure able to predict math performance irrespective of the math achievement measure analyzed, the contributions of the subscales brought out inconsistent results, with the numerical processing anxiety factor being involved in children's math grades and in their ability to fluently solve simple calculations as measured through standardized tests, whereas the situational and performance anxiety factor was only associated with teachers' math assessments. Given that the global MA as well as the numerical processing and situational and performance anxiety factors are equally reliable and valid, the selection of broad versus narrow measures of MA should rely on the specific targeted aim of study.

Finally, concerning the contribution of MA to math performance in the younger compared to the older schoolchildren, we found a significant interaction between global MA and math grades and between the situational and performance anxiety factor and math grades. In both cases, the direction of the associations was the same for the younger children compared to the older children; that is, the children reporting higher levels of MA tended to obtain lower grades, but these associations proved weaker for the older children. This result could be taken as a sign of inadequacy of this instrument at these grade levels. However, due to the exponential cognitive development experienced in children during the school years [78], which implies improvements of WM [79], metacognition [80], and efficiency in the math strategies used [81], it might be likely that math performance was more affected by MA in younger children, whereas their older peers could benefit from their more developed cognitive resources to buffer the detrimental effect of MA on math outcomes. It is worth highlighting that the regressions that involved the numerical processing anxiety factor to predict Math fluency subtest scores did not lose relative prediction power in the older children compared to the younger children. To our understanding, this fact provides support for the adequacy of the SEMA to study MA in the earlier and upper elementary school years.

## Conclusions

In this study, we developed a European-Spanish version of the SEMA [3]. The psychometric indices obtained, and the resulting factor structure revealed that the version of the SEMA

developed is a reliable and valid measure to evaluate MA in children from 3rd to 6th grade. The results found in this study provide meaningful insights into the nature of children's MA in the elementary school years. The association between MA and general anxiety proved the overlap between these constructs, but their differential role in contributing to math performance supported the notion that they are different entities. The exploration of gender differences indicated that the girls reported higher levels of MA than the boys, but the effect sizes were small and vanished away when trait anxiety was controlled for. Differences across grades were found for global MA and the numerical processing anxiety factor but not for the situational and performance anxiety factor. Finally, MA was negatively associated with children's math performance, although the strength of associations varied with the MA measure selected, the kind of math performance analyzed, and the school stage considered. In summary, our findings highlight the relevance of MA in elementary school and the need for an early identification of students at risk of suffering MA to palliate the negative consequences of MA in children's cognitive and academic development.

## Supporting information

**S1 Data.**
(SAV)

**S1 File.**
(DOCX)

## Acknowledgments

We thank the participating children, their parents, teachers, and the administrative authorities of the schools for their cooperation. This research was supported by grant PSI2017-84556-P from the Spanish Ministry of Economy, Industry, and Competitiveness (FEDER funds) and PID2019-107857GA-I00 (FEDER funds) from the Spanish Ministry of Science and Innovation.

## Author Contributions

**Conceptualization:** Noelia Sánchez-Pérez, Luis J. Fuentes, Carmen González-Salinas.

**Formal analysis:** Noelia Sánchez-Pérez.

**Funding acquisition:** Noelia Sánchez-Pérez, Luis J. Fuentes.

**Investigation:** Luis J. Fuentes.

**Methodology:** Noelia Sánchez-Pérez, Luis J. Fuentes, Carmen González-Salinas.

**Writing – original draft:** Noelia Sánchez-Pérez, Luis J. Fuentes, Carmen González-Salinas.

**Writing – review & editing:** Noelia Sánchez-Pérez, Luis J. Fuentes, Carmen González-Salinas.

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
