## [Decision Letter · Decision Letter 0]

23 Feb 2021

PONE-D-20-32824

Assessing math anxiety in elementary schoolchildren through a Spanish version of the scale for early mathematics anxiety

Dear Dr. R Sánchez-Pérez,

We apologize for the delay in the review process of your revised manuscript.

We have now received three reviews on your manuscript PONE-D-20-32824 entitled **“**Assessing math anxiety in elementary schoolchildren through a Spanish version of the scale for early mathematics anxiety** ". **After careful consideration, we feel that it has merit but does not fully meet PLOS ONE’s publication criteria as it currently stands. Therefore, we invite you to submit a revised version of the manuscript that addresses the points raised during the review process. When preparing your revised manuscript, you are asked to carefully consider the reviewers’ comments which are attached, and submit a list of responses to the comments.  Your list of responses should be uploaded as a file in addition to your revised manuscript.

Please submit your revised manuscript no later than 45 days from today. If you will need more time than this to complete your revisions, please reply to this message or contact the journal office at plosone@plos.org. Please include the following items when submitting your revised manuscript:

We look forward to receiving your revised manuscript.

Kind regards,

Bing Hiong Ngu, Ph.D.

Academic Editor

PLOS ONE

Journal Requirements:

Reviewers' comments:

Reviewer's Responses to Questions

**Comments to the Author**

1. Is the manuscript technically sound, and do the data support the conclusions?

Reviewer #1: Yes

Reviewer #2: Yes

Reviewer #3: Yes

2. Has the statistical analysis been performed appropriately and rigorously? 

Reviewer #1: Yes

Reviewer #2: Yes

Reviewer #3: No

3. Have the authors made all data underlying the findings in their manuscript fully available?

Reviewer #1: Yes

Reviewer #2: No

Reviewer #3: No

4. Is the manuscript presented in an intelligible fashion and written in standard English?

Reviewer #1: Yes

Reviewer #2: Yes

Reviewer #3: Yes

5. Review Comments to the Author

Reviewer #1: I appreciate that the author did a rigorous and detailed statistical anlaysis (e.g., ANCOVA, CFA) to address the research questions. However, I have a major concern regarding the novelty of the study. To be honest, I don't see much novelty for this study merely looking at math anxiety, trait anxiety, math performance, and non-verbal IQ. Simply translating SEMA to Spanish contributes little to the extant research.

Also, I am pretty concerned about some conclusions the author drew. For example, under "Math anxiety and math performance", the author stated "the Situational and Performance factor negatively predicted students’ grades 423

given by teachers..." I don't think this could actually "predict", given the lack of evidence of causality.

Reviewer #2: I enjoyed reading this manuscript. Overall, the study provides useful insight into maths anxiety in a young Spanish population. I particularly like the range of maths attainment measures, especially (as the authors point out), studies tend not to include such a wide range. The usefulness of this is emphasised by the regression findings where different patterns of results are shown. The manuscript reads well and I believe the work will add sufficiently to the literature in this field. However, there are several grammatical/wording errors throughout and a thorough proof read to ensure correct English is required before publication. I have made a few minor comments on individual sections below.

Intro

“decreasing levels of experienced anxiety 82 when facing mathematics operations throughout childhood would be expected due to the development of executive functioning”. I am not sure of the logic here given that difficulty level is increased throughout schooling, in line with such development.

Participants

Rather than specifying exact salary, it would help to put the salary breakdown into context, e.g. referring to percentages below, at, or above the mean/median national salary.

Results

The T-tests on gender are not needed alongside reporting the main effect of gender from the ANOVAs.

pp 295-297 – it would help to be more explicit that “total score” refers to total maths anxiety score.

Discussion

In the subsection “maths anxiety and its relation to general anxiety”, to further support the argument for differentiating trait anxiety from maths anxiety, it might be worth highlighting how trait anxiety was no longer a significant predictor in the regression models (on performance) when maths anxiety was included.

Reviewer #3: Overview

This research study examined a Spanish translation of the SEMA, an English math anxiety measure for children, and examined how it related to trait anxiety, grade level, gender, and math grades and performance. Results suggest a relation with trait anxiety, that there are gender differences in math anxiety (when trait anxiety is not covaried), that math anxiety is higher in third graders than in some of the older grades, and that math anxiety is related to math grades and performance, though different subscales are more related for different outcomes. I think this study was interesting and well written. My main concern is the use of a math anxiety measure that has math problems in the questions when using the measure across development. This makes it very difficult to understand the reason behind grade level differences. In addition, I think more information is needed about the items on the scale and the reasoning for removing any items.

Introduction

- line 111: Can you talk more about what issues need to be thought about to make sure that a scale is developmentally appropriate for children?

Method

- line 142: Can you mention what percentage of families answered the socioeconomic and education-related questions?

- line 158: Including math problems in math anxiety questions when doing research either longitudinally or cross-sectionally across ages can cause problems due to the relative ease or difficulty of the math problems for children of different ages. You end up bringing this up later on, but to me this is the largest issue with this measure (an issue with the original SEMA as well, not just your version). This is especially a problem because one entire sub scale has math problems, and the other one doesn’t. So differences in findings between the subscales are difficult to interpret as well.

- line 160-161: Please report the words and/or pictures that made up the response scale. I know the faces they use in their picture scale and I find them to be difficult to interpret – did you test that children understood the pictures well? Were both the words and pictures available to children when they responded or just the pictures?

Results

- line 204: Can you please include a table that lists all of the items and their means, standard deviations, skewness, and kurtosis? And maybe means and standard deviations by grade level? I think including the original SEMA item as well as the Spanish translation would be most transparent for all readers.

- line 204-208: I’d like to know more about the items that you found to be problematic and what information you used to determine that they had low variability – was there a particular cut off? Did these items correlate with other items on the scale?

- line 221 (and a few other places): You report p < .000, which is not possible, probably needs to be p < .001.

- line 222-223: You report the partial eta squared, which is useful, but for the gender difference can you report d as well so that is easier to compare with other studies? (same on line 238). You have a large sample size so I am guessing these ds are quite small.

- line 265 and 268: Can you include a range of values for the factor loadings? Were there any that were low (< .3, or .4)?

- line 269: Was the 2-factor model a better fitting model than the one-factor model? I typically use the Satorra-Bentler statistic in MPlus, I am not sure if you can get this in R.

- Table 5: Because grade is a categorical variable with 4 categories, it is probably more appropriate to dummy code it in the regression analyses than to treat it as a continuous variable.

- Table 5: How is gender coded in the models? I am asking because it is a statistically significant predictor and I don’t know if that means boys or girls had higher math scores.

- line 350: To test whether the relations are different by grade level, a regression analysis with an interaction term between grade and math anxiety would be able to statistically compare if the size of the relations across grades.

Discussion

- line 377 paragraph: The grade issue here – though you discuss it, it is impossible to tease apart why you found these grade differences, and, if anything, the fact that you found them for the subscale that has math problems and not the other subscale lends some support to the explanation that it is because the items have math problems, not because of any developmental differences between students in different grade levels. In addition, this issue arises when interpreting the relations between math anxiety and performance for younger compared to older students. It is likely that what is being measured in younger students is slightly different from what is being measured in older students given the relative ease of the items for the older students. Thus, it does not surprise me that the relation is stronger in younger students where you can really see how they feel in response to more difficult math. Again, this difference in relations may be due to the math problems used in the items, not to actual differences in the underlying construct of math anxiety, and it makes it difficult to interpret this finding. It might help to run the analysis with the two subscales separately and see if the relations are weaker for older children only for the numerical processing factor and not for the situational and performance factor. Regardless, I think that using this scale across multiple ages may be in appropriate given the content of the scale, and it might be best to remove any grade analyses from the paper given difficulty in interpreting these findings and analyses may need to just be run within grade where we can be more confident that the responses mean the same thing.

- line 393 paragraph: Can you talk about how the size of the gender difference you found compares to that found in other studies? And why the gender difference may have gone away when general anxiety was covaried out?

- line 461: Also mention here that the gender differences were not apparent when general anxiety was covaried out.

6. PLOS authors have the option to publish the peer review history of their article (what does this mean?). If published, this will include your full peer review and any attached files.

Reviewer #1: No

Reviewer #2: **Yes: **Dr Thomas E Hunt

Reviewer #3: No

---

## [Author Response · Author response to Decision Letter 0]

19 Apr 2021

April 15th, 2021

Dear editor,

We are very grateful for having the opportunity to revise our manuscript entitled “Assessing math anxiety in elementary schoolchildren through a Spanish version of the Scale for Early Mathematics Anxiety (SEMA)”. In the revised manuscript, we have carefully considered the reviewers’ comments and suggestions and have also taken the chance to improve the quality of the paper. As instructed, we have replied to each comment raised by the three reviewers (the reviewer’s comments have been kept in black ink whereas our answers are in blue color), and uploaded also the revised manuscript in two versions: one marked-up copy highlighting the changes made to the original version and another unmarked version of the revised paper without tracked changes.

The reviewers’ comments were very helpful, and we thank them for their constructive and generous feedback on our submission. After addressing all reviewers’ concerns, we feel the quality of the paper is much improved and suitable for publication in Plos One journal.

Best regards,

Dr. Noelia Sánchez-Pérez

First of all, we have to say that after we received the Editor’s decision letter some missed data from children that participated in the study were found. This has led us to have to re-analyze all data including those collected from the new children. For that reason, the sample size has changed slightly. Fortunately, the results, although with minor changes in some decimal places, remain overall as in the original version we submitted to the journal. Specifically, the results that have changed referred to:

• ANCOVAs analyses with trait anxiety as a covariate (lines 312-329, page 14; the lines and pages correspond to the unmarked version of the revised paper without tracked changes.). In the previous version, there were marginal differences in global MA, numerical processing, and the situational and performance anxiety factors. With the changes in the sample those marginal effects were no significant. 

• The hierarchical regressions to predict math test scores (calculation, and math fluency) with numerical processing anxiety factor as a IV (lines 439-448, table 6, page 20). In the previous version, numerical processing anxiety factor was a significant predictor of calculation and math fluency. Although it remains as a significant predictor for math fluency, its contribution to calculation is not significant in the new version of the manuscript.

Reviewer #1

 I appreciate that the author did a rigorous and detailed statistical analysis (e.g., ANCOVA, CFA) to address the research questions. However, I have a major concern regarding the novelty of the study. To be honest, I don't see much novelty for this study merely looking at math anxiety, trait anxiety, math performance, and non-verbal IQ. Simply translating SEMA to Spanish contributes little to the extant research.

We thank the reviewer for taking the time to review the paper. We are sorry for not being able to attract his/her interest. Our main purpose was to develop an instrument to identify math anxiety among elementary school students in Spain, and to study its psychometric properties. In undertaking this task, the following issues concerning the study of MA in children have been addressed: the measurement of MA in children; the structure of MA at this age period; age and gender differences; the association of MA with anxiety trait; and MA relations with mathematics performance. However, more than a simple translation of the SEMA into Spanish, we aimed to explore the usefulness of this instrument to predict math performance from a broader perspective that includes both standardized math tests and teachers’ scores, as well as in a range of ages beyond those previously explored. As it will be described later on in our reply to the other reviewers’ concerns, from our point of view our results make an important contribution in this respect. 

The new version of the manuscript addresses more seriously the adequacy of the SEMA to measure MA in upper elementary school students by comparing the reliability as well as the structural validity of the instrument in older children compared to younger ones, as well as identifying the contribution of the SEMA scales in the prediction of math performance in older compared to younger students. Another issue more profoundly studied concerns the factorial structure of MA as measured through the SEMA; we have paid deeper attention to the fact that the numerical processing anxiety and situational and performance anxiety factors are highly correlated, and have tested the adequacy of a unifactorial model versus a two-factor model by comparing the respective fit indices provided for the whole sample and splitting the sample by boys/girls, and younger/older children. Additionally, the appropriateness of using a broad measure of MA versus more specific ones is considered in connection with math performance. Finally, and following the other reviewers’ suggestions, we have introduced a great deal of changes in the different sections of the paper that hopefully have improved the quality and the interest of the manuscript.

Also, I am pretty concerned about some conclusions the author drew. For example, under "Math anxiety and math performance", the author stated "the Situational and Performance factor negatively predicted students’ grades 423

given by teachers..." I don't think this could actually "predict", given the lack of evidence of causality.

To make the predictivity issue clearer to the readers we have introduced the specific time/month when the data were collected in the procedure (line 262-267, page 11; the lines and pages correspond to the unmarked version of the revised paper without tracked changes). Specifically, math anxiety scales were collected between October and December 2018, achievement standardized tests between January and Mach 2019, whereas children’s math grades were informed in June 2019. Although we acknowledge causality is a common issue in this sort of studies, we think we are allowed to use the term “predict” in a temporal sense because math anxiety was measured at the beginning of the academic year whereas the math performance measures were obtained later. 

Reviewer #2

I enjoyed reading this manuscript. Overall, the study provides useful insight into maths anxiety in a young Spanish population. I particularly like the range of maths attainment measures, especially (as the authors point out), studies tend not to include such a wide range. The usefulness of this is emphasized by the regression findings where different patterns of results are shown. The manuscript reads well and I believe the work will add sufficiently to the literature in this field. However, there are several grammatical/wording errors throughout and a thorough proof read to ensure correct English is required before publication. I have made a few minor comments on individual sections below.

We thank the reviewer for his/her kind comments about the manuscript. We apologize about the grammatical/wording errors. They have been corrected in the new version of the paper.

Intro

“decreasing levels of experienced anxiety 82 when facing mathematics operations throughout childhood would be expected due to the development of executive functioning”. I am not sure of the logic here given that difficulty level is increased throughout schooling, in line with such development.

We agree that our expectation about how MA would develop throughout school years was rather naïve, bearing in mind that MA can be affected by a diversity of endogenous and exogenous factors, and not only developmental ones. On another side, as it was pointed out by the third reviewer, changes in the scores across grades of global MA as well as the Numerical Processing factor are difficult to interpret because of the different difficulty that specific arithmetic problems involve for students of so different grade levels. For this reason, although we have kept the ANOVAS involving grade to inform about how the scores of the SEMA scales change with grade, we have decided not to fathom deeply on the interpretation of such changes. Nonetheless, we have introduced this issue referring to the work by Sorvo et al., (2017), who attributed the decreasing mean levels of MA as grade increased to the students’ familiarization with the study of math (see lines 93-95, page 4; the lines and pages correspond to the unmarked version of the revised paper without tracked changes).

Participants

Rather than specifying exact salary, it would help to put the salary breakdown into context, e.g. referring to percentages below, at, or above the mean/median national salary.

In order to put the income level into context, we have introduced the mean monthly family incomes of Region of Murcia, the geographical area in which the study is placed (lines 214-215, page 9). It was not possible to express the information as requested by the reviewer because the questionnaire administered did not ask the families to exactly inform about their monthly incomes but rather asked them to choose one among 6 options that expressed different incomes intervals. 

Results

The T-tests on gender are not needed alongside reporting the main effect of gender from the ANOVAs.

We agree with the reviewer, so we have eliminated the t-test analyses, but kept the information about the mean and standard deviation for boys and girls. 

pp 295-297 – it would help to be more explicit that “total score” refers to total maths anxiety score.

We agree that the term “total score” is very unspecific. Instead, we now use the term “global MA” to refer to the total math anxiety score.

Discussion

In the subsection “maths anxiety and its relation to general anxiety”, to further support the argument for differentiating trait anxiety from maths anxiety, it might be worth highlighting how trait anxiety was no longer a significant predictor in the regression models (on performance) when maths anxiety was included.

We thank the reviewer for highlighting this important issue. We have introduced this fact to stress that these two constructs, although correlated, are separate entities (lines 556-558, page 26).

Reviewer #3

My main concern is the use of a math anxiety measure that has math problems in the questions when using the measure across development. This makes it very difficult to understand the reason behind grade level differences. In addition, I think more information is needed about the items on the scale and the reasoning for removing any items.

We are aware that the items of the Numerical Processing anxiety scale contain math problems and operations that are learned in early elementary school and that these problems involve different levels of difficulty to younger compared to older children. However, our position is that this scale, although not suitable to compare the mean levels of MA across grades, can be useful to detect MA in children in the upper elementary years. We based this assumption on research carried out in adult population in which the stimuli (and items in questionnaires) used to uncover math anxious individuals are simple arithmetic operations, and cognitively undemanding tasks (see lines 179-183, page 8; the lines and pages correspond to the unmarked version of the revised paper without tracked changes).

We have decided to keep the ANOVAS involving grade because we believe that showing how the scores of the different SEMA scales vary with grade are informative of the functioning of the SEMA in the sample studied. However, we have warned the reader about the limitation of Numerical Processing to identify true differences in MA across grades. (see lines 523-526, page 25).

We have also provided more information about the reason why we removed items 2, 4, 6, 12, and 13, as addressed below.

Introduction

- line 111: Can you talk more about what issues need to be thought about to make sure that a scale is developmentally appropriate for children?

Following the excellent review by Ganley and McGraw (2016), we have added a new paragraph describing the aspects that should be taken into consideration when generating instruments directed to children. We thank the reviewer for this suggestion because in the new version of the manuscript readers could better appreciate the efforts involved in the development of such measurement instruments (lines 123-139, page 6).

Method

- line 142: Can you mention what percentage of families answered the socioeconomic and education-related questions?

Following the suggestion of the reviewer, we have introduced the percentage of families that answered the socioeconomic and education-related questions (lines 199-211, page 9).

- line 158: Including math problems in math anxiety questions when doing research either longitudinally or cross-sectionally across ages can cause problems due to the relative ease or difficulty of the math problems for children of different ages. You end up bringing this up later on, but to me this is the largest issue with this measure (an issue with the original SEMA as well, not just your version). This is especially a problem because one entire sub scale has math problems, and the other one doesn’t. So differences in findings between the subscales are difficult to interpret as well.

We know that the SEMA was designed for early elementary school students, but we wanted to test its usefulness in upper elementary school too. For that reason, besides the general analyses run with the whole sample, we have run additional analyses in which the sample was divided into younger (3er, 4th grade), versus older (5th, 6th grade) children. For instance, the factorial structure of the SEMA was tested for younger and older children. We also considered the relative predictive power of MA on math performance depending on the school stage. On the other hand, the analyses have been run for the global MA, and for the specific subscales, which bring the reader the possibility to examine its functioning either when the subscales are considered grouped together or isolated. In our opinion, taken together, the results found in our sample support the use of the SEMA as a useful instrument to measure MA in children throughout elementary school years, taking in mind the limitation already mentioned given by the fact that one of the subscales included specific math problems.

- line 160-161: Please report the words and/or pictures that made up the response scale.

We have provided the specific words used for each response option and informed that each option was illustrated with a cartoon face, that has been reproduced with the permission of the original authors in the SF1. 

 I know the faces they use in their picture scale and I find them to be difficult to interpret – did you test that children understood the pictures well? Were both the words and pictures available to children when they responded or just the pictures? 

As introduced in (line 263, page 11), the SEMA was administered in an one-to-one interview and the children had the chance to ask if they did not understand any aspect concerning the items meaning or the scale. In addressing this issue, we have talked to the members of our staff who administered the questionnaire and they told us that children in general answered with no hesitation, giving the impression that they clearly understood the instructions given.

Results

- line 204: Can you please include a table that lists all of the items and their means, standard deviations, skewness, and kurtosis? And maybe means and standard deviations by grade level? I think including the original SEMA item as well as the Spanish translation would be most transparent for all readers.

Following the reviewer’ suggestion, we have introduced the required details in a new table (see Table 1, page 15).

- line 204-208: I’d like to know more about the items that you found to be problematic and what information you used to determine that they had low variability – was there a particular cut off? Did these items correlate with other items on the scale?

In (lines 270-284, page 12) we have clarified our decision of removing items 2, 4, 6, 12, and 13 based on the distribution of scores.

Regarding the correlation of each removed item, we have computed the item-test correlation and the Cronbach’s alpha with and without them for younger and older students:

Younger students

Item Item-test correlation Cronbach’s alpha of the final scale Cronbach’s alpha of the scale with the item

4 .34 .68 .70

6 .22 .68 .68

12 .28 .74 .74

Older students

2 .43 .70 .72

4 .40 .70 .72

6 .24 .70 .70

12 .44 .78 .79

13 .37 .78 .79

- line 221 (and a few other places): You report p < .000, which is not possible, probably needs to be p < .001.

Sure, we have inserted this modification when necessary, throughout the text.

- line 222-223: You report the partial eta squared, which is useful, but for the gender difference can you report d as well so that is easier to compare with other studies? (same on line 238). You have a large sample size so I am guessing these ds are quite small.

Reviewer 2 pointed out that the t-tests on gender were not needed alongside reporting the main effect of gender from the ANOVAs; so, we have eliminated the t-test analyses (and consequently the possibility to report d). However, to help the readers to understand the meaning of partial eta squared we have introduced the following sentences: “The results indicated that grade and sex explain 2.5 and 0.7% of the SEMA scores, respectively” (lines 290-291, page 12); “The results indicated that grade and sex explain 5.7 and 0.7% of the numerical processing anxiety factor” (lines 299-300, page 13); and “The results revealed that gender explain 0.5% of the subscale scores” (lines 309-310, page 13).

- line 265 and 268: Can you include a range of values for the factor loadings? Were there any that were low (< .3, or .4)?

The values for the factor loading ranged from .47 to .63, and from .39 to .65, for Numerical processing Anxiety and Situational and Performance Anxiety scales respectively (lines 369-370, page 17).

- line 269: Was the 2-factor model a better fitting model than the one-factor model? I typically use the Satorra-Bentler statistic in MPlus, I am not sure if you can get this in R.

We thank the reviewer for the suggestion of using Satorra-Bentler statistic. The Package ‘SBSDiff’ (the RStudio package used to apply the Satorra-Bentler test) specifies that this test only runs using the estimator maximum likelihood (ML) and our estimator is "ULS" for unweighted least squares. Our decision to choose the ULS estimator is based on the paper published by Forero, Maydeu-Olivares, and Gallardo-Pujol (2009), where the authors found that ULS provides more accurate results for ordinal variables than the other estimators. Moreover, we have studied other statistics to compare fitting models in RStudio (such as BIC and AIC), but we found the same obstacle: they only work with ML. As a result, we used the fit indices (CFI, TLI, RMSEA, and SRMR) to decide which model (one- versus two- factor) show a better adjustment. 

Attending to the fit indices brought out for the different models, both could be plausible. In the new version of the manuscript, we have paid more attention to the adequacy of broad versus narrow measures of MA in connection with math performance. This issue has been raised in the Introduction (lines 160-171, page 7), and the Discussion section (lines 505-518, pages 24-25, and lines 564-603, pages 27-28).

- Table 5: Because grade is a categorical variable with 4 categories, it is probably more appropriate to dummy code it in the regression analyses than to treat it as a continuous variable.

Following the reviewers’ suggestion, grade was treated as a categorical-ordinal variable. 

- Table 5: How is gender coded in the models? I am asking because it is a statistically significant predictor and I don’t know if that means boys or girls had higher math scores.

Gender was coded as 0 for girls, and 1 for boys. This information has been provided in line 431, page 19.

- line 350: To test whether the relations are different by grade level, a regression analysis with an interaction term between grade and math anxiety would be able to statistically compare if the size of the relations across grades.

We thank the reviewer for the interesting suggestion. Following the recommendation, we have introduced the interaction term analyses for school stage and math anxiety (lines 460-485, pages 22-23), with significant effects between school stage and global MA to predict math grades, and also between school stage and the Situational and Performance factor to predict math grades. The new analyses indicate that higher scores on math anxiety are associated with lower math grades for younger and older students; but this association is stronger for younger than for older students.

Discussion

- line 377 paragraph: The grade issue here – though you discuss it, it is impossible to tease apart why you found these grade differences, and, if anything, the fact that you found them for the subscale that has math problems and not the other subscale lends some support to the explanation that it is because the items have math problems, not because of any developmental differences between students in different grade levels. In addition, this issue arises when interpreting the relations between math anxiety and performance for younger compared to older students. It is likely that what is being measured in younger students is slightly different from what is being measured in older students given the relative ease of the items for the older students. Thus, it does not surprise me that the relation is stronger in younger students where you can really see how they feel in response to more difficult math. Again, this difference in relations may be due to the math problems used in the items, not to actual differences in the underlying construct of math anxiety, and it makes it difficult to interpret this finding.

We thank the reviewer for this caution. As said before, we have highlighted this limitation in the new version of the manuscript.

It might help to run the analysis with the two subscales separately and see if the relations are weaker for older children only for the numerical processing factor and not for the situational and performance factor. 

As has been mentioned, the suggested analyses were introduced in the results section (lines 459-486, pages 22-23), and discussed later at lines 604-621, page 28-29. 

Regardless, I think that using this scale across multiple ages may be in appropriate given the content of the scale, and it might be best to remove any grade analyses from the paper given difficulty in interpreting these findings and analyses may need to just be run within grade where we can be more confident that the responses mean the same thing.

As it was said before, we have decided to keep the grade differences analyses because of its informative value, with caution about the interpretation of the results. Although we haven’t run all the analyses within grade, following your previous suggestion, we have divided the sample into younger (3er and 4th grade), and older (5th and 6th grade) children to ascertain the predictive power of MA on math performance depending on the school stage. Introducing the MA*school stage interaction has brought out interesting results: The predictive power of MA on math performance only proved significant for both global MA and Situational and Performance factor, so that we found a weaker association between MA and math achievement in older children compared to younger ones. Importantly, the association of Numerical Processing anxiety with Fluency standardized test did not vary across school stage (please, see Results, lines 473-477, page 23). This latter result suggests that the Numerical Processing anxiety factor is useful to assess MA in early as well as upper elementary school, which extends the usefulness of the SEMA beyond the original age range.

- line 393 paragraph: Can you talk about how the size of the gender difference you found compares to that found in other studies? And why the gender difference may have gone away when general anxiety was covaried out?

- line 461: Also mention here that the gender differences were not apparent when general anxiety was covaried out.

In the new analyses run comparing average levels of MA for boys and girls, once we covaried trait anxiety, all the previous differences were gone. In the Discussion section (lines 534-546, pages 25-26), we have referred to the effect sizes of the differences as small, which disappeared when general anxiety was controlled, similarly to the study by Ganley and McGraw (2016), and explained this result based on the mediational study developed by Szczygiel (2020), in which it was proved a mediational pathway from gender to MA via general anxiety.

---

## [Decision Letter · Decision Letter 1]

23 Jun 2021

PONE-D-20-32824R1

Assessing math anxiety in elementary schoolchildren through a Spanish version of the Scale for Early Mathematics Anxiety (SEMA)

PLOS ONE

Dear Dr. Noelia R Sánchez-Pérez,

We apologize for the delay in the review process of your revised manuscript. We now have received feedback from two reviewers who have reviewed the revised manuscript PONE-D-20-32824R1 "Assessing math anxiety in elementary schoolchildren through a Spanish version of the Scale for Early Mathematics Anxiety (SEMA)." Based on the advice received plus my own reading, we determined that your manuscript requires another revision before we can consider it for publication in PLOS ONE. Please revise your manuscript as recommended by the reviewer.

Please submit your revised manuscript no later than 30 days from today. If you will need more time than this to complete your revisions, please reply to this message or contact the journal office at plosone@plos.org. Please include the following items when submitting your revised manuscript:

We look forward to receiving your revised manuscript.

Kind regards,

Bing Hiong Ngu, Ph.D.

Academic Editor

PLOS ONE

Journal Requirements:

Reviewers' comments:

Reviewer's Responses to Questions

**Comments to the Author**

1. If the authors have adequately addressed your comments raised in a previous round of review and you feel that this manuscript is now acceptable for publication, you may indicate that here to bypass the “Comments to the Author” section, enter your conflict of interest statement in the “Confidential to Editor” section, and submit your "Accept" recommendation.

Reviewer #2: All comments have been addressed

Reviewer #3: (No Response)

2. Is the manuscript technically sound, and do the data support the conclusions?

Reviewer #2: Yes

Reviewer #3: Yes

3. Has the statistical analysis been performed appropriately and rigorously? 

Reviewer #2: Yes

Reviewer #3: Yes

4. Have the authors made all data underlying the findings in their manuscript fully available?

Reviewer #2: Yes

Reviewer #3: Yes

5. Is the manuscript presented in an intelligible fashion and written in standard English?

Reviewer #2: Yes

Reviewer #3: Yes

6. Review Comments to the Author

Reviewer #2: It is my opinion that the resubmission is a notable improvement on the original submission. My comments have been adequately addressed and I believe the additional changes have resulted in a strong paper. Please note a few minor things to consider prior to publication, should the paper be accepted:

In line with the most recent APA guidelines, presumably all in-text citations involving more than two authors should be changed to “first author et al.”?

Lines 131-132 - “(2) Formulation of the rating scale so that it is 132 clearly understood by children.” needs to be reworded.

Line 679 – “CLARK-CARTER” is unnecessarily capitalised.

Line 357 – I would remove the word “usually”.

Reviewer #3: In general, I believe the authors have addressed the comments from my previous review. Below I outline some continued concerns and suggestions.

- paragraph starting line 270: It would be useful to see the item wording for the items that were removed due to having low variability. Perhaps still include them in Table 1? I wondered if you could talk more conceptually about why they might have had such low rates of agreement of anxiety - were these especially easy math problems, for example? I’m still not sure I agree with removing them unless it was part of your goal to develop a scale with fewer items, which was not stated prior to this. They all seem to be correlated with the other items suggesting they do measure the same construct, but are just picking up on the kids with really high levels of math anxiety. (item 6 is perhaps borderline – it seems to have a pretty low item-total correlation at both age groups).

- line 288: You say “analyses” – can you say if this was ANOVAs or something else and mention what was included in the model before going into the model results?

- line 295: I think you could include the d here for the gender difference (and in the places where there are gender differences reported). The d is not tied to the t-test and can be calculated on its own based on means and standard deviations.

- Table 1: Can you make it more clear which subscale each item belongs to? (even just a note at the bottom of the table would help)

- line 384: I’m not really sure what to take away from your CFA analyses – would you argue that a 1-factor or 2-factor model is what researchers should use? Or that both are reasonable to use? Is a correlation of .84 between 2 subscales of the same construct reasonable? How does this compare to the original SEMA?

- paragraph starting line 406: I found this paragraph a bit confusing to follow because you hadn’t told me what was in each model before you told me results and you talked about the second model (with the covariates and math anxiety) before talking about the first model (with just the covariates), which was sort of backwards order to how the models were run. It also was not made clear that the F, p, and R2ajusted reported were for the whole model – it was written as though these were part of the results just for math anxiety.

- line 426: Why was grade a potential covariate tested for these models but not for the models predicting math grades? In my previous review I suggested treating grade as a categorical variable instead of a continuous variable, but it doesn’t look like you did that in the tables reported in this section as grade would need to be parsed into 3 pairwise comparisons using dummy coding.

- line 446: Mention that numerical processing anxiety did not predict calculation skills.

- line 448: It is unclear if your results in the regression analyses for the 2 factors are different from the zero-order correlations because they are highly correlated with one another or because the covariates are included in the model. Can you add a model where just the 2 math anxiety factors are predictors without the covariates? This would help to tease apart whether it’s including the subscales together or if its’ including the covariates that leads some of them to not be significant despite significant zero-order correlations for both.

- 466: Did the interaction analyses include the same covariates as were previously included?

- line 519: Can you say more about what you think your CFA results mean for the construct of math anxiety? Are these separate components meaningfully different?

- line 538: You mention that the gender difference in global MA and numerical processing anxiety went away, but didn’t it go away for situational/performance anxiety as well?

- line 584: The way this sentence is written implies that numerical processing anxiety predicted both of the standardized math test outcomes, but it only predicted one of them.

7. PLOS authors have the option to publish the peer review history of their article (what does this mean?). If published, this will include your full peer review and any attached files.

Reviewer #2: **Yes: **Thomas E. Hunt

Reviewer #3: No

---

## [Author Response · Author response to Decision Letter 1]

19 Jul 2021

Dear editor,

We are very grateful for having given another opportunity to revise our manuscript entitled “Assessing math anxiety in elementary schoolchildren through a Spanish version of the Scale for Early Mathematics Anxiety (SEMA)” and to improve the quality of the paper. In this revised version, we have carefully considered the reviewers’ comments and suggestions. As instructed, we have replied to each comment raised by the two reviewers (the reviewer’s comments have been kept in black ink whereas our answers are in blue color), and also uploaded the revised manuscript in two versions: one marked-up copy highlighting the changes made to the original version and another unmarked version of the revised paper without tracked changes.

The reviewers’ comments were very specific and detailed, and we thank them for their time. After addressing all reviewers’ concerns, we feel of the paper has improved significantly and we hope it can reach the quality standards for publication in Plos One journal.

Best regards,

Dr. Noelia Sánchez-Pérez

Reviewer #2

1. In line with the most recent APA guidelines, presumably all in-text citations involving more than two authors should be changed to “first author et al.”?

We thank the reviewer for the reminder about the changes in the 7th edition of the APA Publication Manual. Following its guidelines, the in-text citations have been changed.

2. Lines 131-132 - “(2) Formulation of the rating scale so that it is 132 clearly understood by children.” needs to be reworded.

We reworded the sentence this way “use of a rating scale whose options have a clear meaning to children”.

3. Line 679 – “CLARK-CARTER” is unnecessarily capitalised.

The error was amended. 

4. Line 357 – I would remove the word “usually”.

Removed.

Reviewer #3

1. paragraph starting line 270: It would be useful to see the item wording for the items that were removed due to having low variability. Perhaps still include them in Table 1? I wondered if you could talk more conceptually about why they might have had such low rates of agreement of anxiety - were these especially easy math problems, for example? I’m still not sure I agree with removing them unless it was part of your goal to develop a scale with fewer items, which was not stated prior to this. They all seem to be correlated with the other items suggesting they do measure the same construct, but are just picking up on the kids with really high levels of math anxiety. (item 6 is perhaps borderline – it seems to have a pretty low item-total correlation at both age groups). 

The removed items have been included in Table 1 (page 15-16) and marked with asterisks. We have referred to the content of the items as possible explanations for the floor effect found and have supported our decision of not including these items in further analyses on the work by Terwee et al., 2007, in which the absence of items with ceiling or floor effects is considered one of the quality criteria for the measurement properties of health status questionnaires (page 12, lines 280-292 from unmarked version of the revised paper without tracked changes).

2. line 288: You say “analyses” – can you say if this was ANOVAs or something else and mention what was included in the model before going into the model results?

We have clarified the expression in the text by introducing “ANOVAs analyses” (page 13, line 297).

3. line 295: I think you could include the d here for the gender difference (and in the places where there are gender differences reported). The d is not tied to the t-test and can be calculated on its own based on means and standard deviations.

We thank the reviewer for the suggestion. We have calculated the Cohen's d and introduced the results in the reported gender differences (page 13, line 304-305; 315-316; 322-323; 340-341).

4. Table 1: Can you make it more clear which subscale each item belongs to? (even just a note at the bottom of the table would help)

We have specified the composition of two scales by introducing the sentence “Original items were obtained from Wu et al., (2012): items 1 to 10 constitute the Numerical Processing Anxiety factor; items 11 to 20 belong to the Situational and Performance Anxiety factor” at the bottom of the Table 1 (page 16)

5. line 384: I’m not really sure what to take away from your CFA analyses – would you argue that a 1-factor or 2-factor model is what researchers should use? Or that both are reasonable to use? Is a correlation of .84 between 2 subscales of the same construct reasonable? How does this compare to the original SEMA?

We thank the reviewer for pointing to the need to further interpret the CFA analyses. In the Discussion section (page 25, lines 539-544), the meaning of the two factors found is referred to and have put in connection with previous literature. The issue concerning the strong correlation between the two factors is addressed in page 26 (lines 548-560). We based our interpretation on the work by Henschel and Roick (2020) in which the high correlation between the factors found in other instruments derived from the MARS is attributed to the fact that the MARS and derived questionnaires tap a unique dimension of math anxiety, that is, the affective component.

6. paragraph starting line 406: I found this paragraph a bit confusing to follow because you hadn’t told me what was in each model before you told me results and you talked about the second model (with the covariates and math anxiety) before talking about the first model (with just the covariates), which was sort of backwards order to how the models were run. It also was not made clear that the F, p, and R2ajusted reported were for the whole model – it was written as though these were part of the results just for math anxiety.

We have reworded the paragraph to make clear the two regressions tests, the two steps and their results (page 19-20, lines 414-432)

7. line 426: Why was grade a potential covariate tested for these models but not for the models predicting math grades? In my previous review I suggested treating grade as a categorical variable instead of a continuous variable, but it doesn’t look like you did that in the tables reported in this section as grade would need to be parsed into 3 pairwise comparisons using dummy coding.

We thank the reviewer for the question and reminder. We have tested the potential effect of grade in children’s math grades and the results brought out grade differences, with 3rd graders obtaining higher scores than 4th graders (page 19, line 406-409). Consequently, grade was introduced as a covariate in further analyses.

8. line 446: Mention that numerical processing anxiety did not predict calculation skills.

It was clarified in the text (page 22, lines 467-468).

9. line 448: It is unclear if your results in the regression analyses for the 2 factors are different from the zero-order correlations because they are highly correlated with one another or because the covariates are included in the model. Can you add a model where just the 2 math anxiety factors are predictors without the covariates? This would help to tease apart whether it’s including the subscales together or if its’ including the covariates that leads some of them to not be significant despite significant zero-order correlations for both.

We have run regression analyses to predict calculation and fluency abilities, and introduced the paragraph with the following results: “To determine whether the weaker relationship between the situational and performance anxiety factors and the standard tests that were observed in the regression models compared with the correlation analyses, were caused by either the correlation between the two factors or the inclusion of the covariates in the model, we performed an additional regression analysis. As shown in Table 7, the numerical processing anxiety factor did yield a significant contribution to Calculation and Math fluency abilities, but the situational and performance anxiety factor did not” (page 22, lines 462; Table 7, page 23) 

10. Did the interaction analyses include the same covariates as were previously included?

Thanks to the author question we realized that we included gender as a covariate for math grades when in fact they were not significantly related. Consequently, we run again the analyses for math grades. The interaction results did not change, as it is shown in page 23, lines 492-517). 

11. line 519: Can you say more about what you think your CFA results mean for the construct of math anxiety? Are these separate components meaningfully different?

As expressed above, this issue has been addressed in the Discussion section, pages 26 and 27.

12. line 538: You mention that the gender difference in global MA and numerical processing anxiety went away, but didn’t it go away for situational/performance anxiety as well?

We thank the reviewer for pointing out our mistake: we have now specified that the gender differences disappeared for global MA and both factors with the sentence “when the general levels of anxiety were controlled for, the differences found for global MA, the numerical processing, and situational and performance anxiety factors disappeared” (page 27, lines 576-578).

13. line 584: The way this sentence is written implies that numerical processing anxiety predicted both of the standardized math test outcomes, but it only predicted one of them.

As the reviewer has noted, numerical processing anxiety predicted fluency scores, but not calculation skills. We have clarified the specific scale in the discussion section (page 29, lines 622-624).

---

## [Editor Report · Decision Letter 2]

26 Jul 2021

Assessing math anxiety in elementary schoolchildren through a Spanish version of the Scale for Early Mathematics Anxiety (SEMA)

PONE-D-20-32824R2

Dear Dr. Noelia Sánchez-Pérez,

We’re pleased to inform you that your manuscript has been judged scientifically suitable for publication and will be formally accepted for publication once it meets all outstanding technical requirements.

Kind regards,

Bing Hiong Ngu, Ph.D.

Academic Editor

PLOS ONE
---

## [Editor Report · Acceptance letter]

28 Jul 2021

PONE-D-20-32824R2 

Assessing math anxiety in elementary schoolchildren through a Spanish version of the Scale for Early Mathematics Anxiety (SEMA) 

Dear Dr. Sánchez-Pérez:

I'm pleased to inform you that your manuscript has been deemed suitable for publication in PLOS ONE. Congratulations! Your manuscript is now with our production department. 

Kind regards, 

on behalf of

Dr. Bing Hiong Ngu 

Academic Editor

PLOS ONE